# Andalusi Defensive Architecture through Martín de Ximena Jurado's Drawings (Mid-17th Century)

Luis José García-Pulido 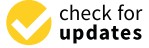

Escuela de Estudios Árabes (EEA), CSIC, 18010 Granada, Spain; luis.garcia@eea.csic.es; Tel.: +34-958222291

**Abstract:** The antiquarian Martín de Ximena Jurado was a pioneer in the historical cartography of the old Kingdom of Jaén (Andalusia, Spain), where he tried to represent emblematic areas with their military defences with his particular graphic language. Not surprisingly, this territory has a high concentration of medieval fortifications. The data and drawings that he made of castles, towers, and defensive enclosures show a special interest in the militarisation of sites and places. He went beyond a simple toponymic study aimed only at finding a correspondence between the ancient name and the location of a settlement based on the evidence provided by coins and inscriptions. The medieval fortifications that he mapped were not drawn in ruins as one would expect they would be in the mid-17th century, but with their most characteristic construction elements. This fact gives it great relevance, as it represents the idealised hypothesis of the state of these constructions at the time of the Castilian conquest in the decades following the Almohad debacle in the Battle of Las Navas de Tolosa (1212).

**Keywords:** medieval fortifications; Almohad and Castilian fortresses; graphic depiction; castle; tower; city walls

## 1. Introduction

Martín de Ximena Jurado was born in 1615 in the eastern part of the countryside of Jaén (south-eastern part of Spain). He was a humanist with ecclesiastical studies in Latin and theology who showed a great interest in history and in the monuments and objects of the past. He belonged to the entourage of scholars who surrounded Cardinal Baltasar Moscoso y Sandoval, Bishop of the Diocese of Jaén, whom he even came to serve as personal secretary (Parejo Delgado 1978, pp. 275–86).

In 1642, he was sent to the town of Arjona to collaborate in the excavations to recover evidence and relics related to the martyrdom to which various saints were subjected. Concerned about the method to be applied, he studied numismatics and palaeography, researching a huge number of documents that were kept in the archives of the area (Castillo Armenteros 2004, pp. 137–45). At the same time, he developed a skill for making simple but intuitive ink drawings. He used graphic representation to capture the features he wished to highlight of the objects, artefacts, structures, architecture, and urban patterns he analysed, producing flat or pseudo-perspective views.

Related to these events, he wrote an account of the discoveries made in Arjona from 1628 to 1642 (Ximena Jurado 1642) including five handmade plans where he drew some of the fortified elements in this municipality (Eslava Galán 1986, pp. 41–44).

In 1643, he completed the *Historia o anales del Municipio Albense Vrgavonense, o villa de Arjona*, a history of the town, where he depicted some of its fortifications (Eslava Galán 1986, pp. 39–46). From then on, he sought to establish fruitful relationships with the intellectuals of the time, definitively opting for historical research work.

The year 1646 marked the beginning of the second stage of his life, which is when he moved with the Cardinal, who had been promoted to Archbishop, to Toledo. He lived there until his death in 1664 and continued to exercise the function of secretary and the post

of ecclesiastical prebendary in the primary cathedral of Spain. The long research activity that he was involved in during his entire life culminated in 1654 with the publication of the *Catálogo de los obispos de las iglesias catedrales de Jaén y Anales eclesiásticos de este obispado* ('Catalogue of the Bishops of the Cathedral Churches of Jaén and Ecclesiastical Annals of this bishopric'), his most important work (Ximena Jurado 1654).

During this second stage, Ximena Jurado developed most of his scientific production, making use of the important libraries and cultural centres in the city. He followed the chronological compilation of data of historical interest, which he interpreted through a wide and varied bibliography (Rodríguez Arévalo 2001, pp. 7–28).

His scientific rigour, remarkable for the time, made him one of the most authoritative historians on the ancient Kingdom of Jaén. He was a pioneer in the cartographic study of this territory, which has the highest density of European medieval fortifications (Cerezo Moreno and Eslava Galán 1989, p. 8; Eslava Galán 1999, p. 17). He tried to represent the defensive architecture of the region with his particular graphic language. This is demonstrated by the vast documentation compiled in the work known as *Antigüedades del Reyno de Jaén* ('Antiquities of the Kingdom of Jaén', *Ms. 1180 B.N.* of the Spanish National Library)[1]. This is an unfinished collection of drawings of various types, personal notes, and a variety of informative notes (Recio Veganzones 1960, pp. 49–68). As can be seen in folio 306v, Ximena Jurado presumably started to write it in 1639, before he was sent to the excavations of antiquities in Arjona in 1642. Around 1646–1647, he must have continued to rework the manuscript shortly after moving permanently to Toledo, although he would have stopped contributing data[2]. It is likely that in the first months of his stay in Toledo, the documentation he had compiled over so many years, in the form of numerous annotations and sketches, was revised. This is indicated by the fact that he took up, ordered, and corrected some works on medals and inscriptions (Mozas Moreno 2018, pp. 72–73).

It is possible that part of the antiquities contained in this manuscript are the result of surveys called *interrogatorios*[3] that Ximena Jurado had requested through Cardinal Moscoso y Sandoval, in order to draw up the history of the Diocese of Jaén. Thus, the manuscript is written in two types of ink: brown for the original writings and drawings, and black for the revisions including erasures, corrections, and new information. In addition, the pages are numbered twice: folios 1 to 133 have foliation and pagination (from 1 to 259), while from 134 to 341, there is only foliation.

In this second block of data, between folios 134 and 176v, we find the set of drawings and sketches relating to the fortifications of cities, towns, and places. In some cases, they are accompanied by measurements, annotations, and brief texts focusing on some specific aspect of the documentation of these sites. Although the section does not have a title, it could be considered as a real chorographic book[4] in the form of a description and catalogue of fortresses of the former Kingdom of Jaén, with a graphic representation of the places inventoried.

The first of these contains a drawing of the fortified town of Baeza in the year 1227, when it was definitively conquered from the Almohad Empire by Ferdinand III of Castile. It had been in Christian hands between 1146 and 1157, and again in 1212, when it was successively won and lost by Kings Alfonso VII and Alfonso VIII. The drawing was made on a folded folio twice the size of the rest of the *Ms. 1180 B.N.*, so it may well be the insertion of a sheet that was finished earlier. In addition, most of the folios (135–176v) contain only a place name, but unfortunately no graphic documents. The dates recorded correspond to 1639 and 1641, that is, to the first years of the manuscript's composition, perhaps before it passed through Arjona. However, these notes for his unfinished graphic chorography were placed on the final pages of the manuscript. This second part of its compilation is centred on data relating to the ancient territory of Jaén, so it also includes a documentary collection made up of fragments of texts transcribed or managed by other contemporary antiquarians or by classical authors including a copy of Aldo Manutio's edition of Antoninus Pius' Itinerary, with the aim of identifying Roman roads in the province of Jaén (Mozas Moreno 2018, pp. 76–77).

On the page 133, which is the last included in the foliation (259), the author inserted a table of 110 places in the bishopric of Jaén as an introduction. He wrote the distance in leagues to each one of them from the capital of Jaén and the number of houses they had, which gives an idea of the existing population in this territory in the middle of the 17th century. After the title, the year 1246 was added in a different typeface in lower-case italics. However, this could not refer to the mid-13th century, shortly after the Christian conquest of these places, as the number of houses indicated would be more appropriate for a date closer to the mid-17th century, so the date could correspond to 1646, when Ximena Jurado was reworking his manuscript in Toledo.

The following are the places for which a drawing of a fortified element was provided in *Ms. 1180 B.N.*: Alcalá la Real (fol. 26), Cazadilla (fol. 41, also in Ximena Jurado 1643, p. 148), Peña de Martos (fol. 63, also in Ximena Jurado 1643, p. 162), Tobaruela (fol. 93), Linares (fols. 109v and 110), Baeza (fol. 134), Alcázar de Baños (fol. 135), Marmolejo (fol. 136), Fuente del Rey (fol. 137), Andújar (fol. 138), Aragonesa o Bretaña (fol. 139), San Julián (fol. 140), La Aldehuela (fol. 141), Benzalá (fol. 142), Cotrufes (fol. 143), Escañuela (fol. 144), and Arjona (in Ximena Jurado 1642, plans 1 to 5 and in Ximena Jurado 1643, pp. 116–18, 121).

In folio 203, a description of the kingdom and bishopric of Jaén in the year 1641 was included (Figure 1). This may have provided a basis for the map of this territory made by the engraver Gregorio Fosman y Medina (Gregorio Forst) in 1653 (Figure 2).

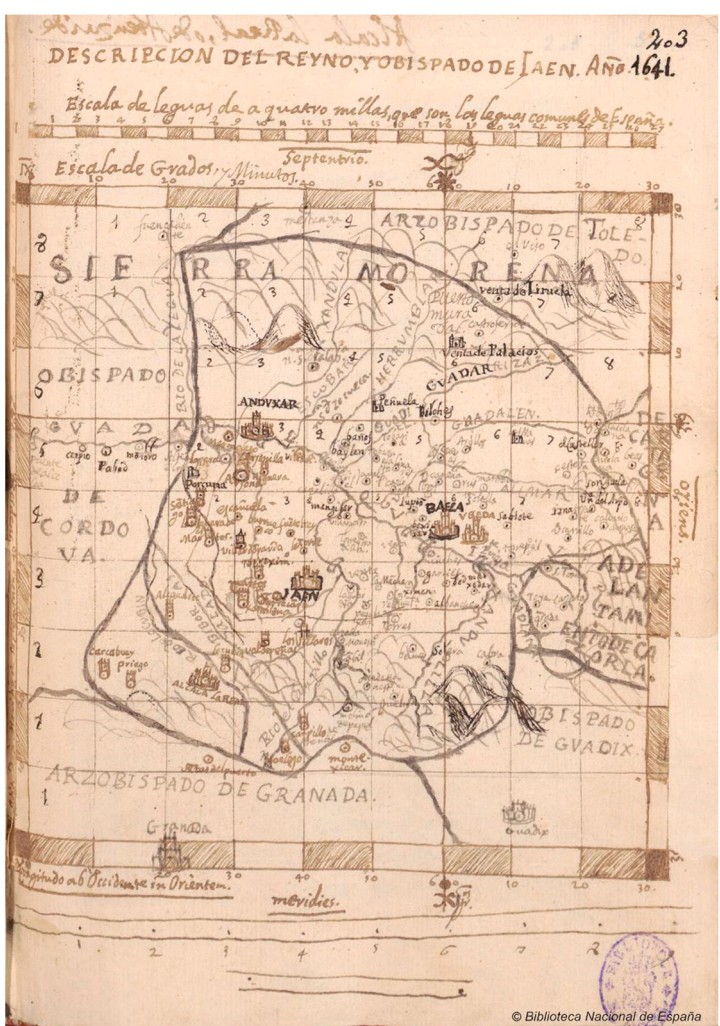

**Figure 1.** *Descripción del Reyno y Obispado de Jaén. Año de 1641* ('Description of the Kingdom and Bishopric of Jaén. Year of 1641'). Map drawn by Martín Ximena Jurado with the places that he was interested in studying. *Ms. 1180 B.N.*, fol. 203.

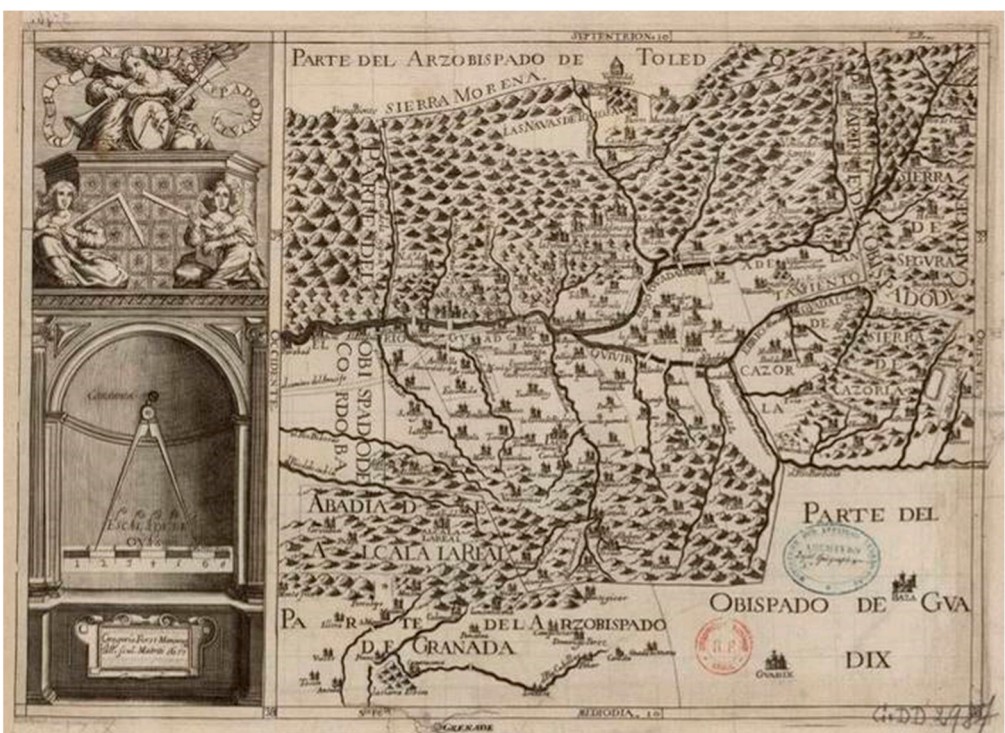

**Figure 2.** *Mapa del Reyno y Obispado de Jaén* ('Map of the Kingdom and Bishopric of Jaén'), engraved by Gregorio Fosman y Medina (Gregorio Forst) to go with the edition of the book *Santos y santuarios del Obispado de Jaén y Baeza* ('Saints and sanctuaries of the Bishopric of Jaén and Baeza') by Francisco Bilches Pedraza (1653) and the *Catálogo de los obispos de las iglesias catedrales de Jaén y Anales eclesiásticos de este obispado* ('Catalogue of the bishops of the churches cathedrals of Jaén and eclesiastic Anals of this bishopric') by Martín Ximena Jurado (1654).

Ximena Jurado relied on the information provided by collaborators, both in terms of numismatics and inscriptions as well as in terms of ancient settlements. In some of them, 'sacred excavations' had been carried out, aimed at locating pre-medieval antiquities that would testify to the grandeur of these archaeological sites before the arrival of the Muslims. However, in his eagerness to document these remains, what he really depicted were the military constructions that had been put in place during the Almohad period, transformed after their defeat at the Battle of Las Navas de Tolosa (1212) by the Castilians.

In the case of the municipality of Andújar, he used Antonio Terrones Robles (late-16th century—ca. 1660) as a referent, whose book *Vida, martirio, traslación y milagros de San Eufrasio, obispo y patrón de Anduxar* ('Life, martyrdom, translation, and miracles of Saint Eufrasio, bishop and patron saint of Anduxar', 1657) had a chapter dedicated to the description of the ruins and buildings of "Los Villares" or "Anduxar el Viejo" (Old Andújar).

For the study of Baeza, the city to which he dedicated one of the most detailed and complete drawings, he would have relied on the contributions of the Jesuit Francisco de Bilches Pedraza (1575–1649), who directed the 'sacred excavations' to look for antiquities and relics in the Alcázar between 1629 and 1640.

For the ruins of Cazlona (Cástulo) Ximena Jurado probably used the drawings in inscriptions made by Gregorio López Pinto, Bishop of Covaleda (17th century). It was likely he who sketched the plan and walled enclosure of this city, which must have corresponded more to the medieval phase (Mozas Moreno 2018, p. 86).

## 2. Results

The thirty or so drawings on fortifications that Ximena Jurado incorporated in *Ms. 1180 B.N.* or published in some of his works can be divided into three main categories: fortified towns and cities, rural castles or small fortified enclosures, and isolated towers.

*2.1. Drawings of the City Walls and Towers of Towns and Cities including Their Castles and/or Citadels*

2.1.1. Alcalá la Real (Figure 3)

Alcalá la Real is depicted in fol. 26 of *Ms. 1180 B.N.* (Mozas Moreno 2018, p. 347). In fols. 163 and 203v, he also wrote the name of the town, but only added the drawing of its coat of arms and a text note on the first of these, leaving the rest of these folios blank.

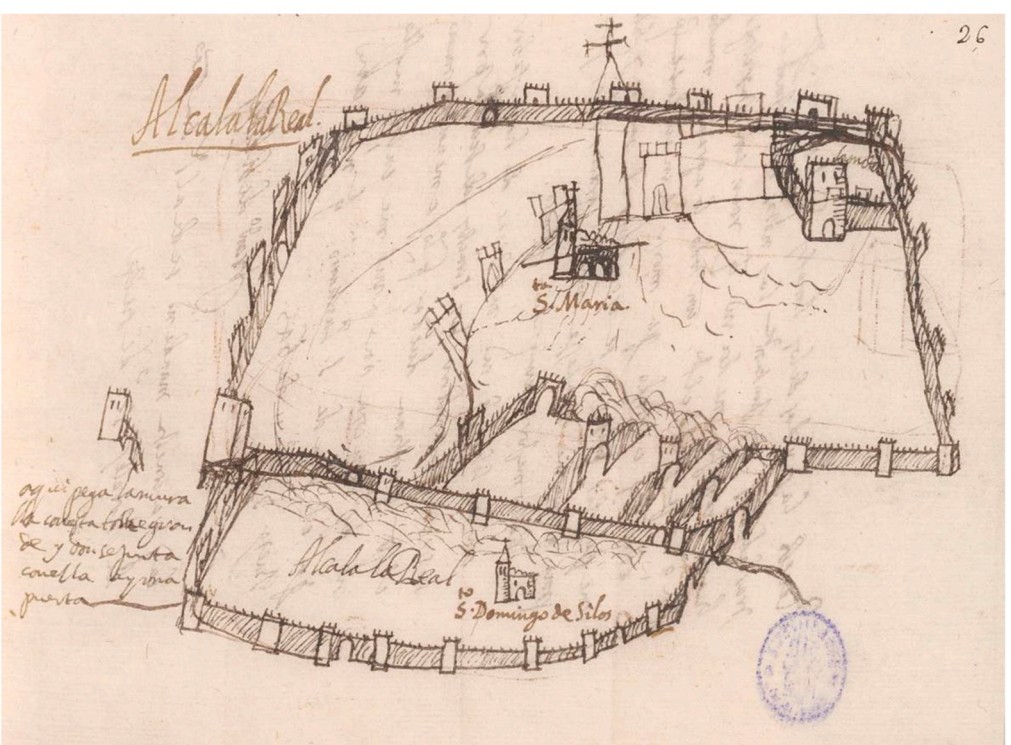

**Figure 3.** Sketch of Alcalá la Real drawn by Martín Ximena Jurado. *Ms. 1180 B.N.*, fol. 26.

On the back of fol. 26, there is text with different spelling signed by Francisco de Bilches Pedraza (1575–1649), the Jesuit in charge of the excavations in the Alcázar of Baeza between 1629 and 1640. Given the format of the paper, which is different from that of most sheets of the manuscript, and the way that it had been folded, not coinciding with the limits of these folios, it might have been a letter sent to Ximena Jurado where he reused the blank side.

Alcalá la Real is labelled and underlined in lighter ink at the head of the page, and three walled enclosures are depicted in black.

The lowest of these contains the name of the town in black ink with the same spelling and writing style as the title. This must have been the only text that accompanied the drawing initially, as all the others are in lighter ink. A church, labelled "Santo Domingo de Silos", was drawn in the lower enclosure. The wall that encloses it from below and from the sides has 10 towers and three gates, while the wall that separates it from the main enclosure of the city has four towers and two gates.

This central enclosure is presided over by the church of "Santa María", which appears with a line running through it, as if it has been crossed out. It is included in an inner enclosure with five wall towers and what appears to be a larger bell tower. Beneath, there are up to four enclosures and just as many gates, two of which are crowned with a tower.

The walled perimeter has 19 towers and three gates. On the left side of the sketch, there is a tower that appears to be free-standing to which a gate is appended, but next to which one can read the following text in red: 'Here the wall meets this large tower and where they join, there is a gate'.

In the upper right-hand corner, the motte fortress is drawn with its main tower and two stretches of wall that connect to the perimeter ramparts.

Folio 163 seems to have been intended to contain information about Alcalá la Real, as is stated in black, but it only contains the drawing of the coat of arms of the city and the following text: 'There are those who write that this city is the one that Ptolemy calls Calicula. But it does not seem to have enough foundation: I noticed in that town when I was there, that it has no trace of anything Roman, but only of Moors'.

On the other hand, in fol. 203v, on the back of the map with the description of the kingdom and bishopric of Jaén from 1641, he also noted "Alcalá la Real, o de Abenzaide", but the sheet remained blank.

Alcalá la Real has the typical layout of a city from al-Andalus: a castle or citadel, an alcazaba or fortified upper quarter, the medina as an extension or continuation of the citadel, and an outer enclosure in the form of a suburb (Eslava Galán 1999, pp. 365–72). The fortified city is located on a hill whose summit is more than 1030 m above sea level, on a plateau covering three hectares, at one of the vital points on the route linking Cordoba with Granada. The natural escarpments, which are very steep, with slopes of almost 100 m above the surrounding territory, contributed to the creation of great defensive elements.

The origin of the walled enclosure can be dated back to around 727, when the fortress was built by Bādīs b. Ḥabūs, later becoming the centre of the Muladi rebellions against Cordoba in 889. The configuration of the defensive enclosure took its final shape during the Nasrid period in the second half of the 13th century, taking advantage of pre-existing fortifications that housed Qal'at Banī Sa'īd, the castle of Benzayde or Abenzaide for the Christians. In 1340, Alfonso XI of Castile laid siege to the city, making it surrender in 1341 after large breaches were inflicted on the outer wall. The Crown of Castile called it the Fortress of Alcalá la Real or Ciudadela de La Mota, and great effort was made to repair and upgrade it due to its strategic position on the main road from Cordoba to Granada (Eslava Galán 1999, pp. 365–72).

Being an archaeological site, material dating back to prehistoric times has been found in the fortress. Excavations have revealed that the intense use of the citadel between the 14th and 16th centuries led to the destruction of the levels of most of the structures from the Andalusi period. Along with numerous elements from the last phase of occupation between the 16th and 17th centuries, a large number of storage structures from the Roman period were found, which may have been used as medieval cisterns or silos, such as those located around the Iglesia Mayor Abacial, built in the 14th century (Moya García 1999a, pp. 132–44; 1999b, pp. 288–300; Calvo Aguilar and Pérez Arjona 2012, pp. 1187–95).

The general layout of this city in al-Andalus was organised into several fortified complexes. The most important enclosure was to the citadel, located on the broad plateau that crowns the motte. The interior space was divided by a wall that ran from north to south, creating a larger space to the west and another to the east, which is where the citadel itself and the Iglesia Mayor Abacial are located today. The citadel is composed of masonry and is the result of repairs carried out after the Castilian conquest. A large part of this complex was surrounded by a first stretch of wall that encircled the rest of the *medina*. It was originally built of rammed earth, but was later covered, and in some cases replaced, by masonry walls. These were reinforced at irregular intervals by towers, some circular and others rectangular.

This would have been the primitive village nucleus of Alcalá and its interior could be accessed via a pathway embedded within the walls that ascended from the north-eastern part of the outskirts of the city to the citadel. It included different gates such as Puerta de Santiago, Puerta de San Bartolomé, Puerta de la Imagen, Puerta de la Plaza, and Puerta del Peso de la Harina.

In addition to the upper enclosure, the progressive increase in population led to the occupation of land on the eastern slope, which made it necessary to enclose other areas with walls of which there are still remains. This first enclosure used to be the outer defences of the medina, and is formed by a large stretch of wall that encircles the south-eastern slope of the motte. Its first layout may have been built around the 11th and 12th centuries in rammed earth or mortar, and covered between the 13th and 14th centuries with thick masonry walls (Castillo Armenteros and Castillo Armenteros 1997).

This other walled enclosure would have protected the so-called Arrabal Viejo or Arrabal de Santo Domingo, located on the eastern slope of the motte under the citadel's protection, acting as the left flank of the main entrance to the fortress.

2.1.2. Martos (Figure 4)

Depicted in fol. 63 of *Ms. 1180 B.N.* (Mozas Moreno 2018, p. 353), there is a very simple sketch of the Rock of Martos seen from the north, crowned by the upper fortress. Fols. 151v and 158 are labelled with the name of the town, but were otherwise left without content.

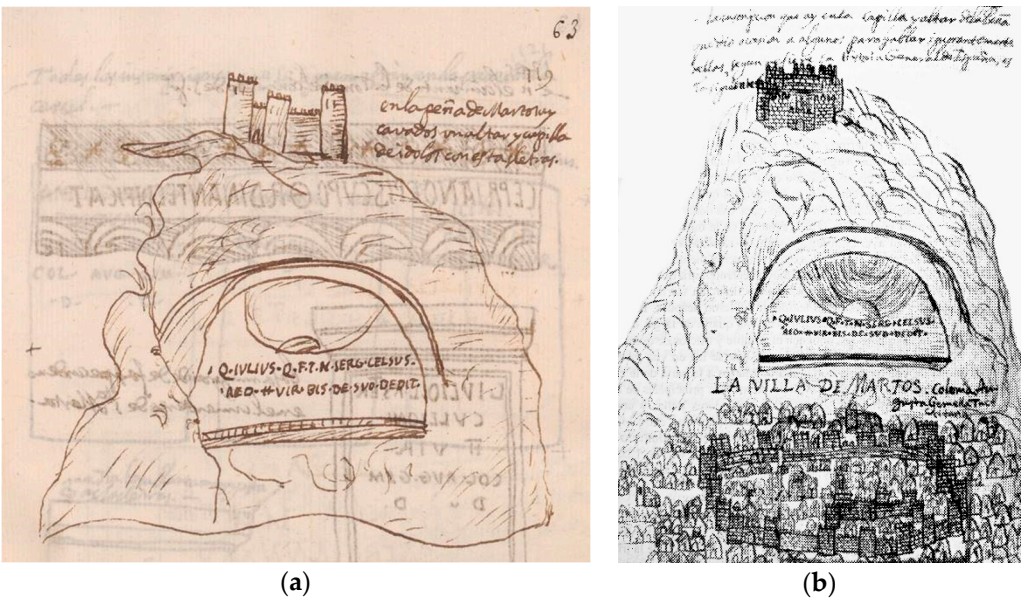

(**a**)　　　　　　　　　　　(**b**)

**Figure 4.** (**a**) Sketch of La Peña de Martos (the Rock of Martos) drawn by Martín Ximena Jurado. *Ms. 1180 B.N.,* fol. 63; (**b**) Another drawing with the city of Martos, published in 1643 in his book about the history of the city of Arjona (Ximena Jurado 1643, p. 162. Arjona Municipal Archive).

In the chapter on Latin inscriptions on stone supports located 'in the district of Martos of the Order of Calatrava, in Martos' (fols. 60–70), the Rock of Martos is represented in a frontal view, with a very simplified image of the northern elevation of the high fortress, with its circular towers and the gate on this flank (fol. 63). In this sketch, however, in the motif is not that of the said fortress, but the existing Latin inscriptions: 'On the Rock of Martos, [there is] an altar and idol chapel carved with these letters' and 'on the corner of a tower in the wall next to the houses of Bernardino de Avoz', already in the urban area. In fol. 70 of *Ms. 1180 B.N.*, he introduced a drawing with the 'arms of Martos, castle of its colour in a field of gold in this form the castle on the rock', which alluded to the iconic image of this rock fortress.

Another drawing of the 'altar and idol chapel' labelled 'The Town of Martos: Colonia Augusta Gemella Tuccitana' was introduced in his book *Anales de la ciudad de Arjona* (Ximena Jurado 1643, p. 162). In this work, they are represented in a pseudo-perspective formed by elevations in a succession of planes containing the towers, walls, and individualised dwellings with gabled roofs of the village of Martos in its lower part as well as the high fortress in the upper part.

There were two fortifications in Martos: the one at the rock was a magnificent castle located at an altitude of more than 1000 m above sea level, and the one in the town around 774 and to the north-west of the aforementioned high promontory.

In the walled enclosure that surrounded Martos, which was oval-shaped in an east–west direction, the low fortress is represented in the form of a towered bastille. Some remains of this fort can still be seen such as the keep, the Almedina (the tower of the city), and the Albarrana[5] as well as sections of the city wall. Although it is not represented, on the other side of the hill, there was another wall that started at the foot of the rock and reached the high fortress.

During the Late Middle Ages, the Command of the Calatrava Military Order controlled this region. They built their main castle on top of La Peña of Martos as a symbol of their power in this region. This fortification had a key role in protecting this territory from attacks from the Nasrid kingdom of Granada during the 13th to the 15th century (García-Pulido et al. 2020).

In the 10th century, the *kūra* of Ŷaīyān (Islamic territory administrated from Jaén) had 16 districts, of which, Tušš (Martos) was one. The Arabic sources give this city the status of *ḥiṣn* (a defensive enclosure to protect one or more settlements), with a strong military role as the main defence of Jaén (Castillo Armenteros and Castillo Armenteros 2003; Castillo Armenteros and Alcázar Hernández 2006, p. 159).

Some scholars have argued that there was already a fortification on La Peña at that time, which would have controlled this territory (Gutiérrez Pérez 2009a, pp. 25, 52; 2013, pp. 11–16). In 1224, Ferdinand III of Castile began the first military incursions into the upper Guadalquivir Valley, supporting the local ruler of Baeza (al-Bayyāsī). In the Navas de Tolosa Treaty (1225) between the two parties, the Castilian king gained Martos. Ferdinand III initially entrusted Count Álvar Pérez de Castro with Martos. The brother of the Almohad Caliph, Abū-l-'Ulā, laid siege to the settlement in 1227 (Olivares Barragán 1992, p. 180). The fortified city withstood the assault, but La Peña fell into the attacker's hands. After these events, at the end of 1228, the Castilian king entrusted the Order of Calatrava with the defence, rule, and organisation of Martos and its surroundings (Gutiérrez Pérez 2009b, pp. 34, 48) including Víboras and Porcuna (Ruiz Fúnez 2010, p. 154).

In 1238, the first ruler of the Nasrid Dynasty, Muḥammad I, laid siege to the fortress. Seven years later, Martos was surrounded by an army from Granada during a raid. However, the intent to take the fortress failed because of the support from the forces sent by Ferdinand III. The city was also the home base of the Castilian conquest of Jaén, which ended in 1246. From them on, the Nasrid border came to lie to the south of the Command of Martos.

After the death of Ferdinand IV of Castile in 1312, a new period of instability began. The Nasrid king Ismaīl I seized the opportunity to launch one of the most intense attacks on the district of Martos. This happened in 1325, after the defeat of a Castilian inroad against Granada. The sultan subjected Martos to an exhausting siege in which gunpowder was used for the first time in this territory (Eslava Galán 1990, pp. 155–56; 1999, pp. 225–30). According to Ibn al-Jaṭīb, the city was taken, raided, and destroyed (Burgos Núñez 1998, pp. 37–45), although the castle of La Peña was not conquered. During the 15th century the pressure from Castile moved the border further south (Castillo Armenteros and Alcázar Hernández 2006, pp. 171, 173), which did not prevent the Nasrid operations against the city during the next few decades. Among the most important ones was the attack by Abū-l-Ḥassān 'Alī's army (Nasrid king from 1464 to 1482 and from 1483 to 1485). All of these assaults forced the Order to adapt their fortresses, until the instability of the border land disappeared after the conquest of Granada in 1492. From then on, the castle of La Peña was gradually abandoned until it fell into ruin (García-Pulido et al. 2020, pp. 1–29; García-Pulido and Navarro Palazón 2023, pp. 145–52).

### 2.1.3. Castle of Linares (Figure 5)

This is sketched twice in fols. 109v and 110 of the *Ms. 1180 B.N.*, the second of which is labelled "Armas de la villa de Linares" ('Arms of the town of Linares') with a lighter ink. This municipality appears on the list of places in the bishopric of Jaén in fols. 133 recto and verso, but he did not dedicate a complete folio to it.

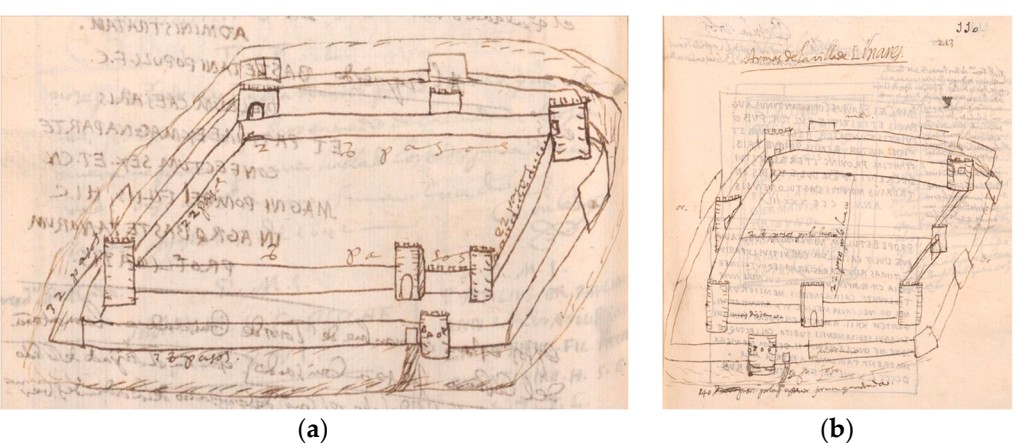

(**a**) (**b**)

**Figure 5.** Two sketches of the castle of Linares drawn by Martín Ximena Jurado. *Ms. 1180 B.N.*, fols. 109v (**a**) and 110 (**b**).

At the end of the chapter that Ximena Jurado dedicated in *Ms. 1180 B.N.* to the antiquities 'in the town of Linares' (fol. 96v) and 'in Cazlona [Cástulo]' (fol. 97), in fol. 109v, he introduced two fragments of inscriptions that would be 'in the door of the ante-wall of the castle of Linares' and 'in the tower of the door of the castle of Linares'. He accompanied them with a pseudo-perspective sketch of the castle, although he did not label the name, as he did in fol. 110 (Eslava Galán 1999, pp. 362–65).

In this first sketch (fol. 109v), he indicated that the inner walled enclosure would have been 28 paces on the north and south sides and 22 on the east and west walls. It would have had five cylindrical crenellated towers, four on the corners with access from the parapet, where battlements are depicted on two sides, and one on the south wall. The crenellated gate-tower would be prismatic and closer to the north-west corner tower. The outer wall would have been 38 paces on the north and south walls and 32 on the east and west, with some cylindrical towers without battlements and a crenellated tower with arrow slits protecting the bridge to the east, which spanned the moat to an open gate in the outer wall. Initially, there may have been an *albarrana* tower on this site, possibly of a different shape, isolated from the whole, until the parapet and moat were added.

In the sketch in fol. 110, he gave the same dimensions: from the eastern (left) to the western (right) front '28 paces through the hollow', and from the northern (bottom) to the southern (top) '22 paces, through the hollow'. Its perimeter was '140 paces on the outside, square in shape'. This castle would have had a wall on which Ximena Jurado drew six cylindrical crenellated towers with access to a raised gate from the parapet, four at the corners, and one in the centre of the west and east sides. The square gate-tower with direct access would have been on the north front, and although he depicted it more or less in the centre of the canvas, he wrote 'more distance' next to the tower on the north-east corner. It would have had an 'ante-wall' that would also have had turrets, with a cylindrical tower without battlements but with arrow slits. It was located on the north side, protecting the bridge that crossed the moat to reach an open gateway over the ante-wall.

Francisco de Torres, in his history of Baeza, which he began around 1633 and finished in 1677, described this fortress as follows (Mozas Moreno 2018, p. 354):

'The castle of Linares is in the middle of the town, very old, strong and well built... It is adorned with six square towers, five round and one square, which rises above the gate facing north with a strong and high crenellated wall, which

makes a spacious courtyard. It has another parapet with some towers, moats and loopholes, where a platform is formed, before entering the castle, and after the parapet, which at least takes more than 1500 fighting men, with a deep and narrow entrance.' (Torres 1677, pp. 214–15, Figure 178)

The enclosure may date from the 13th century, passing into Christian hands after the surrender of Baeza around 1227. They reinforced the fortress with machicolations and the raising of slender towers that protruded over the parapet to give them more autonomy, replacing the previous quadrangular and rammed earth towers with other cylindrical masonry towers. In the 15th century, this fortress took on an active role in the civil wars, which led to the reinforcement of the outer walls. The settlement that gave rise to Linares from Cástulo was formed around the castle.

The circular keep with an upper machicolation in the north-west corner is the only remaining defensive element of this castle, together with some parts of the wall. It was demolished between 1803 and 1804 by order of Francisco María Solano Ortiz de Rosas, Marquis of La Solana and General Captain of Andalusia.

In 2011, an archaeological intervention was carried out that revealed the existence of the remains of a second keep next to this tower, which could be the existing access to the northern interior wall of the castle. The work carried out revealed, next to the tower, the entrance bridge to the fortress as well as a double wall and what could have been a large religious building (Martínez Aguilar 2012).

### 2.1.4. Baeza (Figure 6)

Ximena Jurado dedicated a large drawing to it in fol. 134 (Mozas Moreno 2018, pp. 340, 343), which is larger than the rest of the sheets of *Ms. 1180 B.N.* It contains one of the most elaborate sketches of this manuscript. It is framed by a double line on which it is written the year when Ferdinand III of Castile definitively conquered Baeza. This date seems to have been repainted in black ink, at least the last two digits, a correction that is more visible in the name of the municipality. Ximena Jurado seems to be putting forward a hypothesis of the walled structure of the city at the time of the Christian conquest. Nevertheless, the drawing shows urban elements that are anachronistic for this time such as the defensive elements of the city recently won from the Almohads, together with the main civil and religious buildings and public spaces that would have been built from that time onwards.

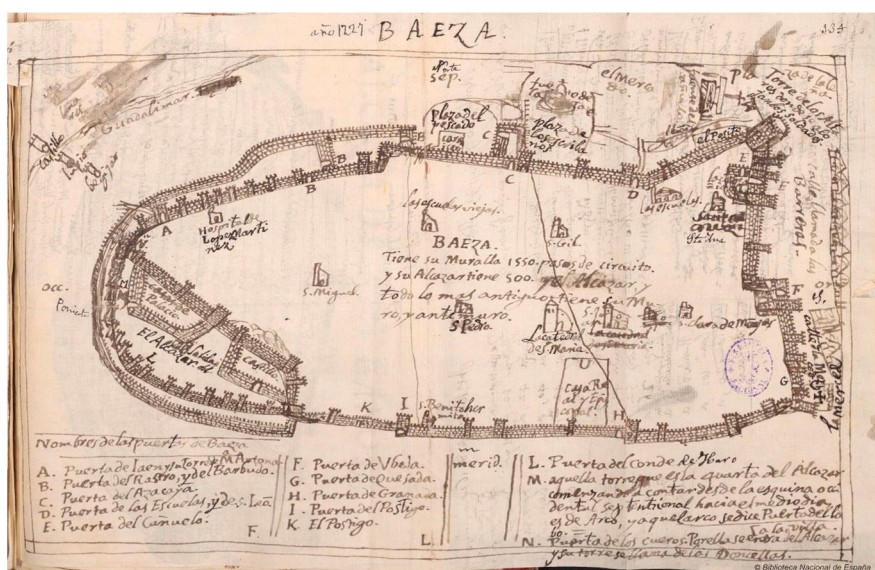

**Figure 6.** Sketch of Baeza drawn by Martín Ximena Jurado. *Ms. 1180 B.N.*, fol. 134.

The drawing is made on a sheet of paper folded in four parts that is twice the size of the rest of the manuscript, so it may well be the insertion of a previously completed sheet,

given that the annotations on the back have nothing to do with the sketch and are arranged differently in each of the quadrants. Furthermore, when this folio was inserted into the manuscript, it was cut out, which affected the text on the back, but not the text on the side on which the sketch is drawn.

The drawing was done in lighter ink, but several texts that had faded more were rewritten in darker ink, possibly not by Ximena Jurado himself. Although the form of the superimposed letters attempts to adapt to the initial writing, in other cases, terms were completed or duplicated with a different spelling. This can be seen in the cardinal points, which the author spelt as usual: "sep. [septentrional]" (northern), "merid. [meridional]" (southern), "occ. [occidental]" (western), "or. [oriental]" (eastern), and rewritten with the words "norte" (north) and "oeste" (west), or the abbreviations "m." [*meridies*] (south) and "es." [este] (east).

The drawing contains an abundance of text, both inside and outside the built spaces and at the foot of or next to the most important buildings. In addition, numbering was introduced with capital letters that refer to a legend with the names of the gates of Baeza. These are as follows:

'A. Jaén Gate and its Tower of María Antonia.

B. Puerta del Rastro and Puerta del Barbudo.

C. Azacaya Gate.

D. Gate of the Schools and San León Gate.

E. Cañuelo Gate.

F. Úbeda Gate.

G. Quesada Gate.

H. Granada Gate.

I. Postigo Gate.

K. El Postigo.

L. Puerta del Conde de Haro.

M. That tower is the fourth tower of the Alcazar starting from the western and northern corner towards the south, it has an arch, and that arch is called Puerta del Lobo.

N. Puerta de los Cueros, one enters through it from the Alcazar to the village, and the tower is called Puerta de las Doncellas'.

Within the walls, at the centre of the image, there is text saying: 'Baeza. Its wall has 1550 paces of circuit, and its Alcázar has 500, and the Alcazar and all the most ancient parts have their walls and antemurals'.

As religious buildings within the walls, "La catedral de Santa María" and the churches of San Miguel, San Pedro, San Gil, San Juan (on an initially crossed-out drawing of the cathedral), Santa Clara de Monjas, Santa Ana (with the name rewritten in a different type of ink and lettering), and the hermitage of San Benito were drawn and labelled. "Las escuelas" [the schools], "las escuelas Viejas" [the old schools], "Hospital de López Martínez", and the space occupied by "Casa Real y Episcopal" were also highlighted.

In the western part, the enclosure of "El Alcázar" was depicted, protected by an outer wall. The enclosure of the castle was drawn standing on some rocks at the highest point, and on the other side of the exit, the palace. Further on, in the upper left-hand corner of the sketch, the icon of a tower represents the towns of Begíjar, Lupión, and, on the other side of the River Guadalimar (with "Bethis" crossed out), is the rewritten word "Castillo", which in the original may have been "Cástulo", together with some words that are unintelligible because they have been smudged with fresh ink, deliberately it seems, and between which vineyards were kept.

Other urban elements outside the walls included the fish, scriveners, and the firewood squares, the butcheries, the so-called "Taza" and "Cañuelo" fountains, the market, the road

to Jaén, and the streets called "Las Barreras" and "La Merced" next to the enclosure of the convent of La Merced, which was marked with a cross. On the other side of this street, a broken row of undifferentiated houses was drawn on the elevation, the only residences represented in the sketch.

With a hexagonal prismatic shape, represented in axonometric view compared to the rest of the elements that are shown in a top–down elevation, the "Tower of the Altars where the soldiers are housed" (known today as Los Aliatares Tower) is drawn in hypertrophy, with a bugle-shaped pennant hanging from a mast hanging out of a window. That this tower has a rectangular prismatic shape has, in fact, nothing to do with Ximena Jurado's detail.

The perimeter of the walls of Baeza was about 2.5 km long, with 53 towers or turrets and another six in the fortress. As for the gates, those at Barbudo, Azacaya, Cañuelo, and Úbeda were double, while the rest were single.

This settlement has a prominent position at the western end of the district of Las Lomas, dominating the Guadalquivir Valley, and the richness of its land has made human settlement possible since prehistoric times. It had precedents from the Bronze Age, in the Iberian period (when it was an important Oretan site), Carthaginian, and Roman times. The Andalusi fortification of Baeza would have had its origin in the works undertaken in 886 by the governor of Jaén. Shortly afterwards, the warlord Umar ibn Ḥafṣūn added the fortress to his domain and held it for almost 20 years.

During the 9th century, the kingdoms of Granada, Seville, and Almería were at war with each other. Alfonso VII of Castile conquered Baeza in 1146, but in 1157, it was taken back by the Almohads, only to be taken two years later by Ibn Mardanīsh of Murcia. During the reign of the Almohad Caliph Yūsuf I (1163–1184), the city must have been fortified as part of his construction programme.

On 21 July 1212, a few days after the Battle of Navas de Tolosa, the Christian army reached Baeza, finding it unguarded and deserted, although it was not taken. No major damage must have been caused to the defences, as they withstood a siege from 1224 to 1225 when a local warlord, al-Bayyāsī, rebelled against the Almohad Caliph al-'Ādil. After this, the citadel finally came into the possession of the Castilians when al-Bayyāsī offered it in return for their help. After his death, the Mudejar population tried to recover it without success, and after the revolt was put down, they were exiled in 1226, when it began to be repopulated with Christian settlers. Fernando I ceded it as a lordship to the knight Sancho Sánchez, who in turn suffered a long siege by the Nasrid king in 1268. The place underwent further sieges in 1407, 1464, and 1475, the latter due to the Castilian Civil War (1351–1369). Within this walled enclosure were the castle-palace of the royal authority and the church, later converted into the collegiate of Santa María del Alcázar.

In 1476, the fortress was demolished on the orders of the Catholic Monarchs so that it would no longer be used as a defence during the struggles within the nobility. At that time, there was a confrontation between the Carvajales families, who were more predominant in Úbeda, and the Benavides, who were dominant in Baeza. Nowadays, only ruins remain and some of the walls have been torn down. The southern front of the wall, adjacent to the fortress, has also disappeared (Eslava Galán 1999, pp. 342–50).

As a result of this destruction, the hill gradually lost settlers, to the point of being almost uninhabited in the 18th century and the collegiate church was transferred to the parish of San Andrés (Rodríguez Molina 1985).

Ximena Jurado drew this sketch in the context of the crisis of the 17th century, which had a particular impact on Baeza, preventing the consolidation of institutions as represented by the cathedral chapter or the university. In addition, many of the city's fortified structures were already in ruins or had disappeared, which is why he would have formulated a graphic hypothesis about the configuration of its defences at the time of the conquest in 1227, or once Christian rule over the city was established, protecting itself against the Nasrids.

2.1.5. Castle of Baños de la Encina (Figure 7)

Profusely drawn on the following page (fol. 135), it is also larger than most of the folios in *Ms. 1180 B.N.* It contains a representation of one of the best preserved fortresses from al-Andalus. It is labelled "Alcazar de Baños" (Mozas Moreno 2018, pp. 340–41) and is accompanied by a rubric. The detailed drawing is framed in a simple rectangle and contains the cardinal orientations "septentrion" (north), "medio día" (south), occidente (west), and "oriente" (east) including only the word "norte" (north) arranged vertically, in a different font and darker ink.

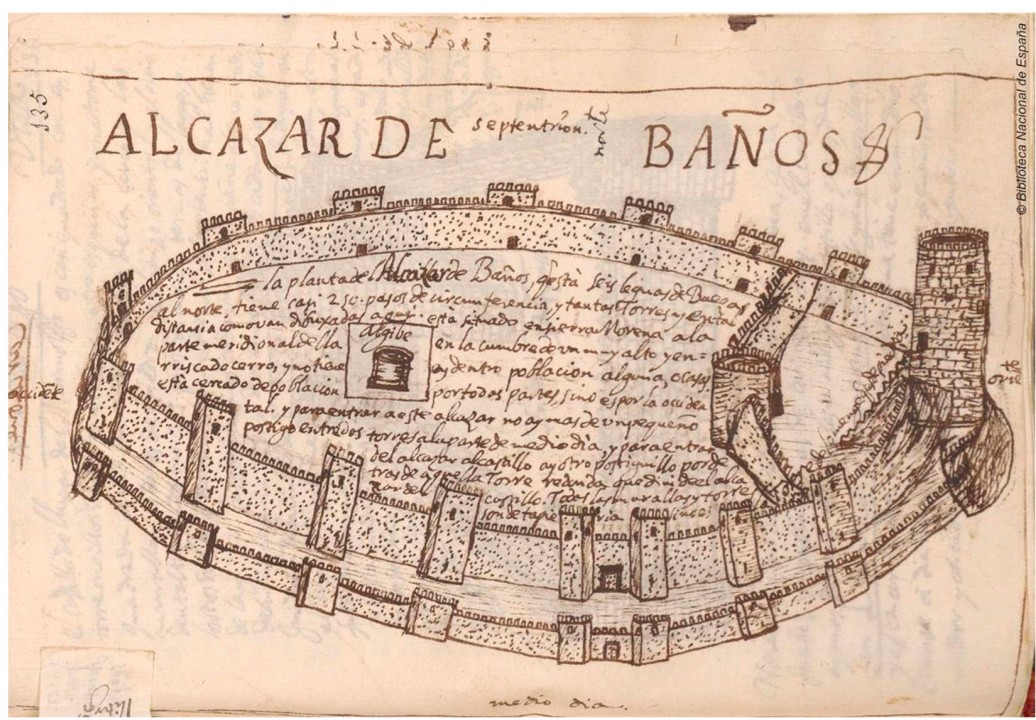

**Figure 7.** Sketch of the castle (Alcázar) of Baños de la Encina drawn by Martín Ximena Jurado. *Ms. 1180 B.N.*, fol. 135.

The interior of the enclosure includes a long handwritten description:

'The ground plan of the Castle ("Alcázar") de Baños, which is six leagues from Baeza to the north, is almost 250 paces in circumference, and has as many towers and in such a distance it preserves as drawn here. It is situated in the Sierra Morena to the south of it, on the summit of a very high and curly hill, and today it has no population or houses within it. It is surrounded by population on all sides, if not on the west. And to enter this fortress there is only a small gate between two towers on the south side, and to enter from the fortress to the castle there is another small gate behind the round tower that divides it from the castle. All the walls and towers are made of rammed earth, except for the keep, which is made of stone'.

Some lines of this text are interrupted when they reach a box containing a perspective drawing of a cylindrical curbstone, above which is labelled "aljibe" (water cistern).

The hill on which the fortress of Burgalimar (from Arabic Burŷ al-ḥammām, Castle of the Baths) stands is at an altitude of 429 m above sea level, from which one can see a vast expanse of land as far as the surrounding mountain ranges. It is located in the extreme south-west of the municipality of Baños de la Encina (Holm Oak Baths).

This fortification was conquered in 1147 by Alfonso VII before passing back to Almohad power in 1157. In 1189, it was taken by the troops of Alfonso VIII of Castile and Alfonso IX of León to be recovered again by the North Africans in 1212 on their way to

the Battle of Navas de Tolosa. Despite its defeat in that confrontation, it remained in their hands until it was definitively conquered by Ferdinand III of Castile in 1225 (Eslava Galán 1999, pp. 357–58).

The castle adapts to the relief of the hill on which it stands, with an irregular ground plan in the shape of a seven-sided parallelepiped. Along its walls are 14 square towers (Ximena Jurado drew 15) with a reduced ground plan and the keep, popularly known in Spanish as the Almena Gorda, has a trapezoidal ground plan and a curved north-east front, although Ximena Jurado drew it in a cylindrical shape.

The gateway has direct access, flanked by two towers, without the defensive bend that characterises Andalusi fortresses. Ximena Jurado drew another in front of it in the antemurium with a similar layout, without being unprofiled.

After the conquest, the Castilians added some elements: a keep with three barbicans on its outer façade, which came to replace the easternmost of the original towers, and an inner division, which came to enclose part of the fortress, with a circular keep that today appears to be severely crumbling, joined to the rest of the fortress by a wall of rammed earth. In Ximena Jurado's depiction, it is crowned with parallelepiped battlements.

The castle of Baños also had an ante-wall or barrier surrounding the ramparts, well represented in Ximena Jurado's drawing. Today, it has almost completely disappeared except for the vestiges that may be preserved inside the houses on Calle de Santa María.

All of the rammed earth work including the walls with its 14 towers with traces of masonry and arrow slits was erected during the Andalusi period, while the masonry part was constructed after the Christian conquest in 1147, being limited to the keep and the inner cylindrical tower. On the ground floor of the keep, the remains of the old Almohad tower can be seen.

The square towers have lost their internal division on the ground plan, although the access openings are at different heights. The lower ones are at the same level as the ground, while the upper ones are at the level of the parapet or walkway.

The entrance to the castle, on the southern flank, is formed by a horseshoe arch, sheltered under another arch of the same type, but slightly pointed. On the inside, there is another pointed arch with its abutment, flanked by two of the square towers.

The walls and towers of the fortress are topped by concrete merlons, erected during one of the restorations in the 1970s. The walls have caissons with heights ranging from 70 to 90 cm, with a predominance of 75 cm (Muñoz-Cobo Rosales 2009, p. 62). They have a false break-up of mock ashlars measuring 0.80 × 2.05 m, with plaster lines of 10 cm. Their dating has been controversial, as, together with the castle of El Vacar in Cordoba, they have been used to justify the claim that this motif was a feature of the Caliphate period in the 10th century (Ferrer Morales 1996, pp. 6–9; Muñoz-Cobo Rosales 2009, pp. 57–106). It was thus argued that the only work of Almohad origin would be the rammed earth wall (Cerezo Moreno and Eslava Galán 1989, p. 74). However, archaeological research has confirmed that the wall and towers containing the mock masonry were also erected at that time (Moya García 2016). Despite the controversial attribution and provenance of an inscription from 967–968, made during the reign of the Umayyad Caliph al-Ḥakam II, decontextualised and transferred to the Spanish National Archaeological Museum in 1907, which has traditionally been linked to the construction of this fortress (Canto García and Rodríguez Casanova 2006, pp. 57–66), the very construction techniques used in the rammed earth walls have led to the main workmanship of Burgalimar being ascribed to Almohad building (Azuar Ruiz and Ferreira Fernandes 2014, p. 401).

2.1.6. Andújar (Figure 8)

Drawn in fol. 138, it is also larger than most of the folios in *Ms. 1180 B.N.* It contains a sketch labelled 'the city of Andújar', and on the right side, again "Anduxar" vertically and in small letters (Mozas Moreno 2018, p. 345).

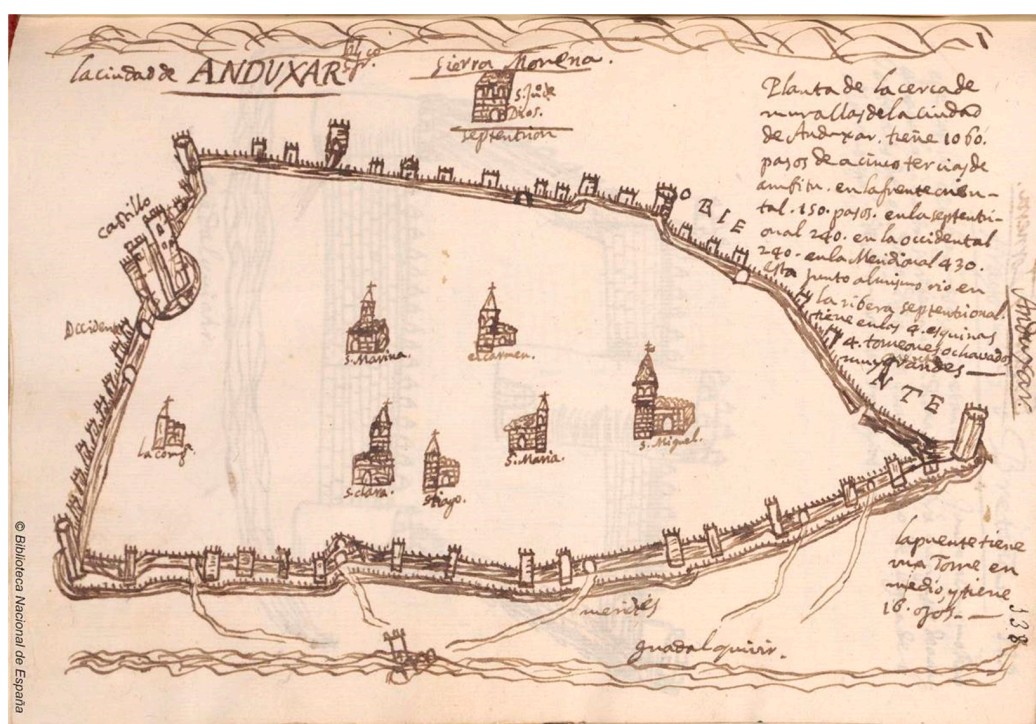

**Figure 8.** Sketch of Andújar drawn by Martín Ximena Jurado. *Ms. 1180 B.N.*, fol. 138.

In this case, the text was inserted in the upper right-hand corner, indicating: 'Plan of the walls of the city of Andújar. It has 1060 paces. On the eastern front, 150 paces. On the northern, 240. On the western, 240. On the southern, 430. It is next to the Guadalquivir River on the northern bank. It has four very large octagonal towers at the four corners'. This text was written after indicating the cardinal points "septentrion", "meridies", "occidente", and "oriente"; as the latter was covered by the text, it was rewritten in capitals with the letters following the line of the wall.

To the south of the walled city, where an ante-wall stood, runs the Guadalquivir River, which could be crossed. Ximena Jurado indicated 'the bridge has a tower in the middle, and has 16 arches', drawing this infrastructure with four arches and a tower between them.

It seems that the walled enclosure of Andújar was built in the Almohad period at the end of the 12th century or very early 13th century, according to a single plan, as it was a strategic town that controlled the passage to the Guadalquivir Valley. The citadel must have been built at the same time.

The perimeter of its walls was trapezoidal in shape. It had as many as 12 gates, but initially, there were no more than seven. The fortification as a whole comprised 48 towers, four more distant octagonal towers facing the river, an *albarrana* tower, an ante-wall, an embankment, and moats. On the side that faced the river, it had a fortress that had completely disappeared in the 17th century. It served to reinforce the walls of the town and controlled the access to the town from the north.

Part of this fence was covered with ashlar masonry in the 15th century. When the city outgrew the enclosure inside the walls, streets were built along the walls, and houses began to be built attached to them.

The Andújar enclosure survived until well into the 19th century, but today, only a meagre 5% of the original perimeter, which was about 1470 m, has been preserved (Eslava Galán 1999, pp. 70–79).

The first schematic map of the city of Cordoba, made in 1752, has a certain similarity to the general layout and content of Ximena Jurado's sketch of Andújar (García Ortega and Gámiz Gordo 2010, pp. 27 (Figure 3) and 31).

### 2.1.7. Arjona (Figure 9)

Fol. 152 of *Ms. 1180 B.N.* was intended to contain information on this municipality, but the folio is empty. However, Ximena Jurado devoted up to nine personalised drawings to the city in other books, from urban views and planimetries to the representation of detailed areas of the town, where excavations had been carried out in search of relics (Ximena Jurado 1643, pp. 3–8).

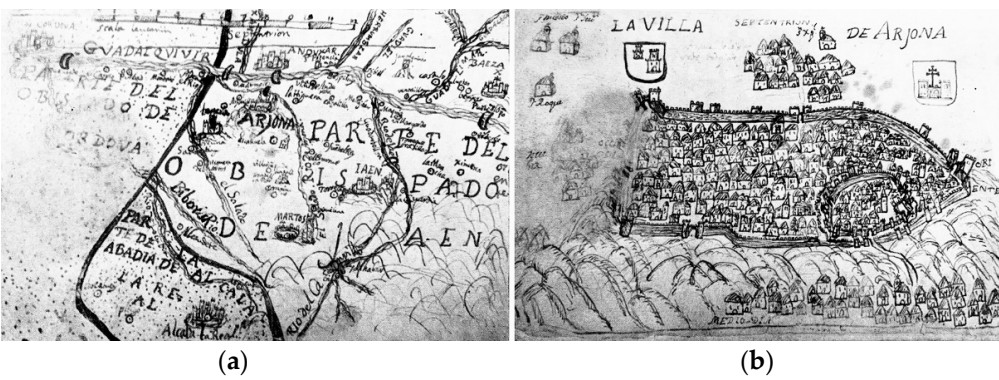

(**a**)　　　　　　　　　　　　　　　　(**b**)

**Figure 9.** Situation of Arjona in the bishopric of Jaén (**a**) and urban view of this town (**b**) drawn by Martín Ximena Jurado for his history of Arjona (Ximena Jurado 1643, pp. 7–8. Arjona Municipal Archive).

Ximena Jurado was sent to Arjona in 1642 to collaborate in the excavations of relics. As a result of this work, around this year and in the following one, he published two books in which the drawings were included. Perhaps for this reason, fol. 152 of the *Ms. 1180 B.N.*, where "Arjona" is labelled, appears empty, which is also the case for the nearby town of Arjonilla (fol. 151). It does include abundant epigraphic, numismatic, and heraldic documentation between fols. 48 and 56, assigning it up to fol. 59 with blank pages on which it was planned to introduce more inscriptions. This is the chapter entitled "en la villa de Arjona, y su Arziprestadgo", where he also includes antiquities located in Arjonilla, Cotrufes, Hardon, and Herrerías. Among them, he drew a 'column of alabaster, which is inside the castle of Arjona in the corner of a tower, which makes an arch to the keep' (fol. 48), another 'at the entrance of the Alcázar in the foundation of the Torre del Ariete' (fol. 49), and two pedestals that 'are inside a cistern, which seems to be the work of the Moors. The vault of the cistern rests on them' (fol. 48v). This water tank is located next to the Church of Santa María del Alcázar, built over the Andalusi mosque, which in turn was built over a primitive late Roman-Visigothic church, constructed over the temple erected in honour of Caesar Augustus. This cistern was probably built in the Almohad period. It is divided into three naves covered with a brick vault, which rest on the pedestals that supported the statues in the Roman temple, one of which is dated to 21 BC.

The most interesting part of his history of the town of Arjona is the one that deals with the period from the 13th century to 1643, with the different chronicles and a series of documents such as privileges and grants awarded to the Council of Arjona (Ximena Jurado 1654). In this book, he structures the study along a chronological line with various news items, with the aim of highlighting the importance of Arjona through the figure of its Christian martyrs, Saint Bonoso and Saint Maximiano, in order to glorify them as agreed at the Council of Trent. Both brothers were natives of Iliturgi, near present-day Mengíbar. They were officers in the Roman army who refused to abandon their faith, for which they were martyred in 308. This news would have been collected in the 'Acts of Martyrdom in Urgavo de la Betica' (Arjona), written around 312 by Saint Felix, Bishop of Guadix, who presided over the Council of Elvira in Granada. They were later sent to Toledo and collected by Flavio Lucio Dextro in the 5th century in his *Cronicón de Omnimoda Historia* or *Historia Universal*, fragments of which were printed in Saragossa in 1619 by Fray Juan de Calderón. They were illustrated in 1627 with notes and commentaries by Licenciado Rodrigo Caro and Fray Francisco de Bivar. This text may have favoured the climate for promoting the

search for skeletal remains in 1629 to which the desired category of religious relics could be attributed.

In his history of the town of Arjona, Ximena Jurado provided an extensive description of the town of Arjona (Ximena Jurado 1643, pp. 3–8) to go with his drawings:

'This place is noble and illustrious for its great antiquity, the greatness of its population, the defence of its walls, [...] chosen as a court by the Moors, the head of a kingdom at one time, the origin and beginning of the House and Kingdom of the Arab Kings of Granada [...].

[...] it is all surrounded by walls and towers, once strong, all made of lime and stone, and now, for the most part, ruined and chipped. The shape of the town is like that of a boat. In the walls there are 24 towers and four gates.

All of which are fortified, each one with two stone towers, very strong with such an art, that one of these towers covers the door in such a way that from the outside it cannot be seen, and thus it is necessary to enter through it with a certain roundabout way, with whose turns they were very safe from the war machines with which the opponents intended to break them when they attacked this town.

The Alcázar is situated on the top of the hill, on the south side; its shape is thus round, because it has only two corners at the two ends of the southern wall, which is somewhat straight.

This Alcázar is surrounded on all sides by walls and ante-walls, with many strong towers along the circumference of the wall; it has twenty towers and in this respect in the ante-wall; it has three gates [...].

At the eastern gate of the Alcázar is the castle with ten towers [...]. It had a moat around this castle for its better defence; its gates were two, both small; one on the inside of the Alcázar, to go out of the castle to the same Alcázar and main square that is located in front of the gate, which looks to the north, and another to go out of the same castle to the town. The circumference of the Alcázar is 1633 rods and that of the castle 267.'

In the surroundings of Arjona, a human presence can be attested to since the Palaeolithic Age by the findings of pieces of stone and human remains, given the possibilities of exploitation presented by the countryside of Jaén. In the Plaza de Santa María in Arjona is located a plateau that corresponds to the first village centre and the highest social prestige of the town from at least the Bronze Age (2000–700 BC), as attested to by the possible discovery of an Argaric burial place in 1628, which was incorrectly thought to be the relics of Christian martyrs (Castro Latorre and Eslava Galán 2022, pp. 86–87, 116–17, 203–4). In the Iberian period, the village was an important *oppidum*, which, together with its neighbours Iliturgi, Isturgi, and Obulco, received privileged status from Caesar after the Battle of Muda. It became one of the first settlements to achieve full Roman citizenship with the title of Municipium Albium Urgabonense.

In the Andalusi period, it became the citadel of the city, with a wall, ante-wall, and fortress. The Andalusian citadel integrated as part of its ante-wall has some stretches of the Iberian wall from the 3rd century BC, with its characteristic sloping form. As a whole, Arjona had one of the most complete sets of urban fortifications on the Iberian Peninsula with three lines of walls, twenty-two towers, two watchtowers, a fortress, a castle, and a cistern. It had an outer walled enclosure that embraced the town, the citadel with the fortress at one end as well as the administrative and commercial quarter with its own walled enclosure and its ante-wall.

In this period, it was called Qal'at Arŷūna, where the Banū Bāhilah settled after the conquest. The city took part in the final battles of the Emirate of Cordoba (9th century), at which time its walls were reinforced.

Tradition has it that on 19 July 1195, on the day of the Battle of Alarcos, the last great Muslim victory on the peninsula, Muḥammad b. Yūsuf b. Naṣr, commonly known as Ibn al-Aḥmar, was born in Arjona. He was to become the founder of the Nasrid Dynasty

(Muḥammad I) and of the last Islamic kingdom on the Iberian Peninsula after settling in Granada, which was more protected and further away from the border, following the Christian conquest of the Guadalquivir Valley after the Battle of Navas de Tolosa (1212). He managed to establish himself as a ruler by using the prestige acquired through his exploits during border battles and the support of the members of his family, the Banū Naṣr, or Banū l-Aḥmar, and of his relatives the Banū Ašqīlūla.

Arjona fell into the hands of Ferdinand III of Castile thanks to a pact with Ibn al-Aḥmar in the spring of 1244, in which houses and inheritance were given to his knights. In 1433, the castle ended up belonging to the Order of Calatrava, while the rest of the citadel belonged to the town council. In the middle of that century, the defences of Arjona underwent successive repairs due to its involvement in the civil war between Juan II and the Infante Don Enrique, who would end up being enthroned as Enrique IV of Castile.

Arjona contains remains of different chronologies and building traditions. Some rests of the primitive Iberian *oppidum* are preserved at Plaza del Mercado, which was extended in Roman times to an enclosure that embraced part of the foothills. After the Muslim conquest, from the year 888 onwards, an intense remodelling would have taken place. The builders of the fortress reused the remains of the old walls and added towers or raised their structures. In 1132, Arjona was considered a strong and safe place, which enabled it to resist subsequent sieges in 1244, 1277, 1316, and 1367.

The work published in 1634 by Ximena Jurado shows that the outer wall was ruined and chipped, that a large part of the keep had collapsed, and that at several places, the only remains left of the Alcázar wall were its foundations. This deterioration escalated from 1639 onwards due to the plundering of its building materials (Eslava Galán 1986, pp. 25–91; 1999, pp. 79–93).

The drawings that Ximena Jurado published in his works show the process of graphic production that he followed, from the taking of quick notes and sketches in *Ms. 1180 B.N.*, to the further elaboration of some of the folios that he inserted folded in that manuscript, to the final presentation of this graphic material in a book.

In the case of Arjona, he made both territorial and urban drawings, which were more detailed for the places where excavations were being carried out.

He drew a territorial map (Ximena Jurado 1643, p. 7) quite similar in layout to the 'Description of the kingdom and bishopric of Jaén' (Figure 1), and depicted with higher detail the north-western quarter of the previous map, in the area where Arjona is located (Figure 9). He also drew an urban view of "La Villa de Arjona" (Ximena Jurado 1643, p. 8). In this drawing, there are also two coats of arms, the details of which he had already sketched in *Ms. 1180 B.N.* This drawing is similar in style to that of the town of Martos, where he resolves this type of view more correctly; with the walls, towers, and gates in pseudo-conical aerial view (a front elevation view with superimposition of planes of depth, although without applying perspectival criteria), and with the western wall erroneously depicted. The walls are not cut into ashlar, but their lines tend to be horizontal. In the interior, the houses are depicted in a very motley manner, with no clear distinction between streets or public spaces. To this end, simple and disproportionate iconic drawings of houses are introduced, with their undifferentiated elevation view, with two windows, a door, and a grated gable roof. This filling resource makes it difficult to identify relevant buildings, although the three churches can be seen by the bell tower crowned by a cross; however, they are not larger than the houses, nor are they labelled. Neighbourhoods outside the walls can also be seen with groups of houses without boundaries, next to the hermitages, recognisable by the belfry on the roof. The citadel is fenced in, with its ante-wall visible as well as the castle and the area of the San Nicolás sanctuary, which is marked with a cross. The mountainous relief he drew towards the south shows that the town is high up and overlooks a hillside.

The latter, in the form of pseudo-axonometric views with abundant explanatory texts, more faithfully show the real state in which the fortifications of Arjona were found.

In addition to these general views, he made five more detailed plans of the relic finds, accompanying the specific book that he wrote as an account of these events (Ximena Jurado 1642, plans 1 to 5), in which he drew fortified elements of the city (Eslava Galán 1986, pp. 41–44).

The first plan is entitled 'Description of the town of Arjona' (Figure 10). It shows a plan view of the circuit of the city walls, with the walls, towers, and gates folded inwards and hierarchised according to their size as well as the enclosure of the citadel and castle in greater detail, also indicating the layout of the outer walls with their profusion of breaks and outlines (Ximena Jurado 1642, plan 1). The most important religious and civil buildings are not indicated. He also referred to the 'Sanctuary where Saints Bonoso and Maximian and many other Christian Saints suffered martyrdom in the Persecutions that the Roman Emperors raised against the Church'.

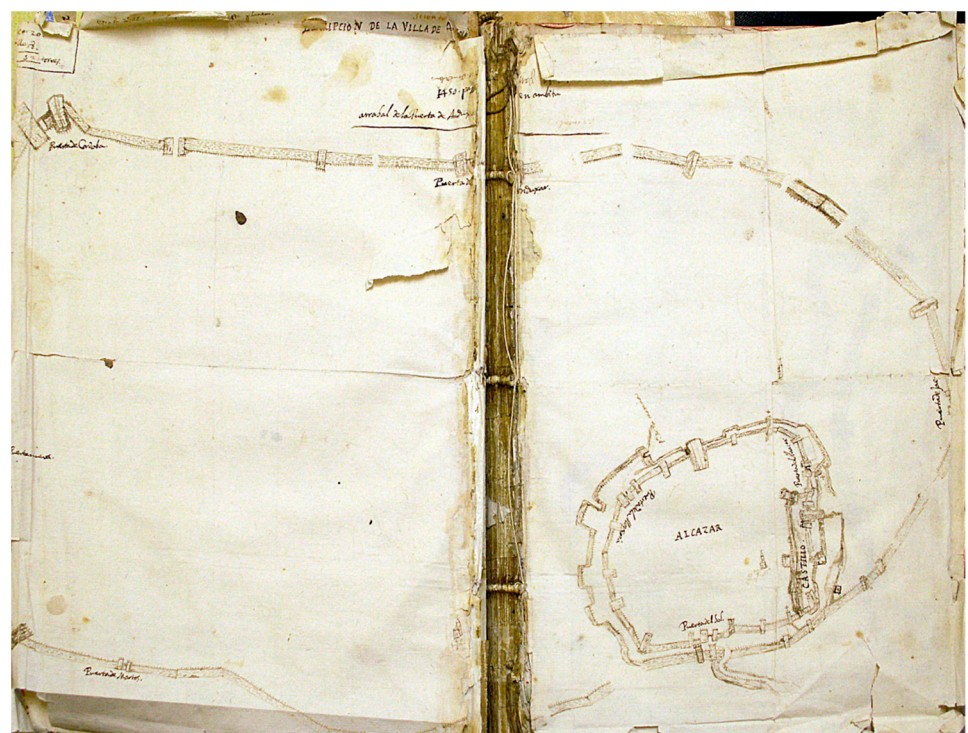

**Figure 10.** Description of the town of Arjona drawn by Martín Ximena Jurado (1642, plan 1). Centro Documental y Biblioteca del Instituto de Estudios Giennenses, Diputación Provincial de Jaén, sign. MAP-B 201-1.

The second plan follows the same drawing criteria, zooming in with greater detail on the area next to the citadel and the castle where the excavations had been carried out (Figure 11). Using a system of icons, the positions of the objects found in the 'moat of the castle' and in the sanctuary of San Nicolás are indicated. In this plan, he introduced a generalised quartering of the masonry of the towers and walls, with the representation of certain details such as some of the battlements (Ximena Jurado 1642, plan 2). In addition to his generalised criterion of marking the four cardinal points with words instead of a compass rose, two simple graphic scales were introduced.

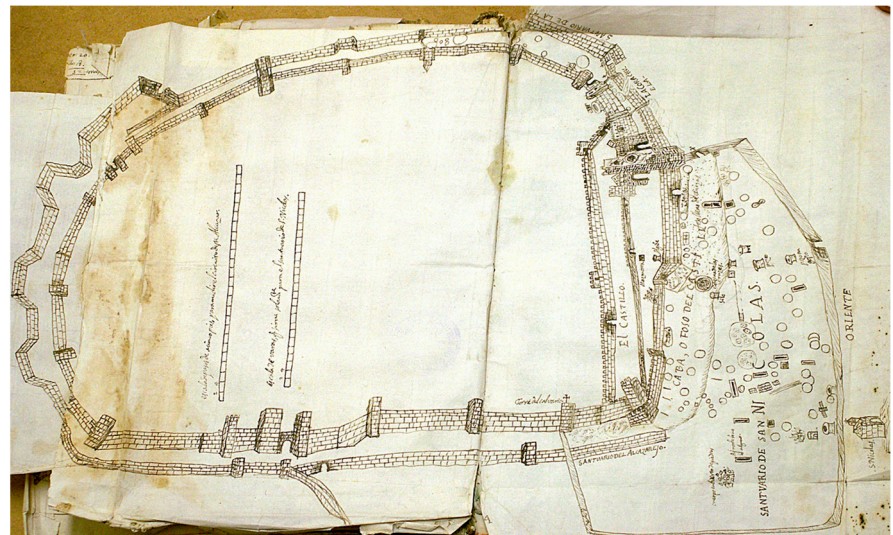

**Figure 11.** The citadel, castle, and the sanctuary of San Nicolás in Arjona drawn by Martín Ximena Jurado (1642, plan 2). Centro Documental y Biblioteca del Instituto de Estudios Giennenses, Diputación Provincial de Jaén, sign. MAP-B 201-2.

The third drawing has a more elaborate graphic scale and a compass (Ximena Jurado 1642, plan 3), which represents in cavalier perspective the area of the findings in the sanctuary of San Nicolás (Figure 12), with references to different towers and other fortifications in this area.

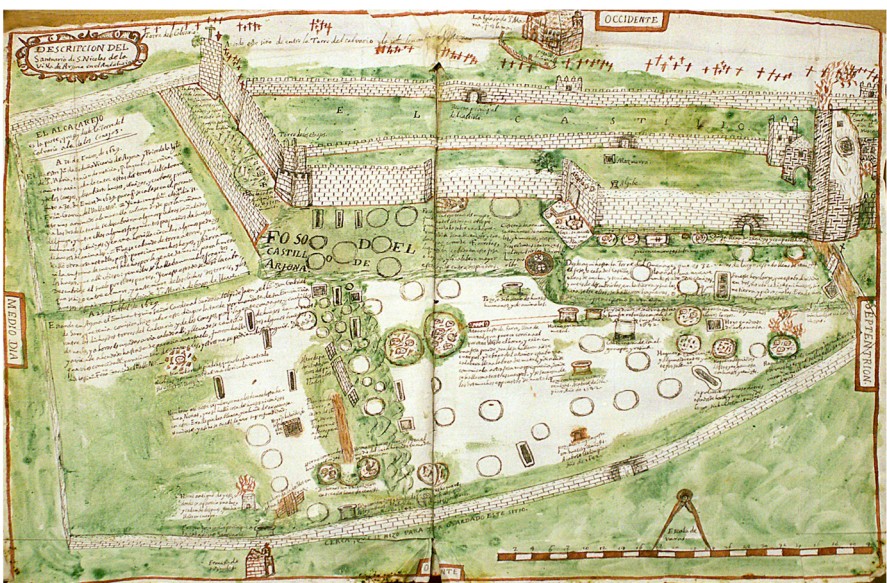

**Figure 12.** Description of the sanctuary of San Nicolás in Arjona drawn by Martín Ximena Jurado (1642, plan 3). Centro Documental y Biblioteca del Instituto de Estudios Giennenses, Diputación Provincial de Jaén, sign. MAP-B 201-3.

The fourth drawing is a 'description of the Sanctuary' (Ximena Jurado 1642, plan 4) with a pseudo-axonometric view, indicating the state of conservation of the towers, gates, and walls with an exploded view of the walls (Figure 13). Thus, it shows where the 'bossed or diamond-shaped' ashlars were found, which he attributed to the ancient period. In addition to the labels and interior texts, he introduced a legend with Latin letters in capital letters.

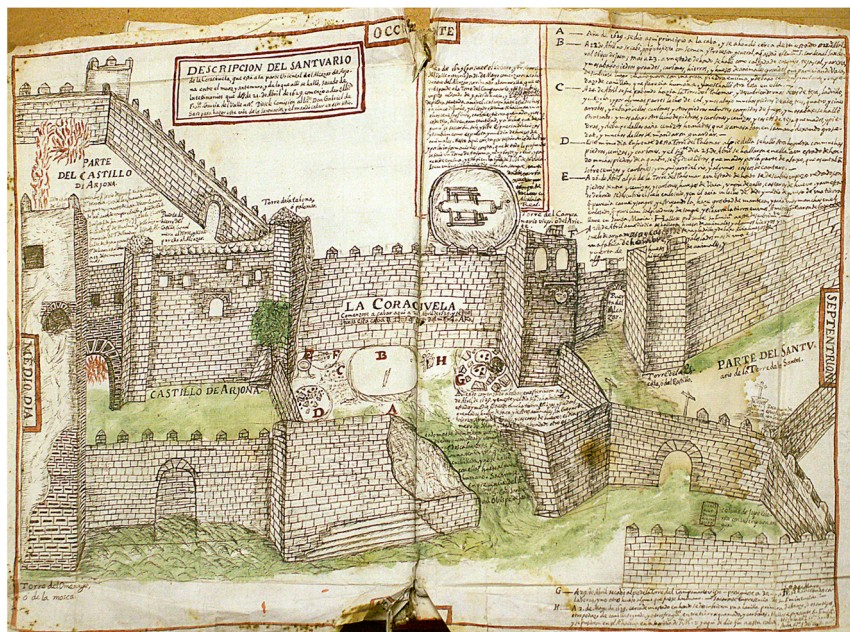

**Figure 13.** The finds in the sanctuary of San Nicolás drawn by Martín Ximena Jurado (1642, plan 4). Centro Documental y Biblioteca del Instituto de Estudios Giennenses, Diputación Provincial de Jaén, sign. MAP-B 201-4.

The fifth drawing is a continuation of the detail of the previous one (Figure 14) towards the 'Sanctuary of the Tower of the Saints' (Ximena Jurado 1642, plan 5), between the wall and the antemurium in pseudo-conic perspective (Eslava Galán 1986, p. 43).

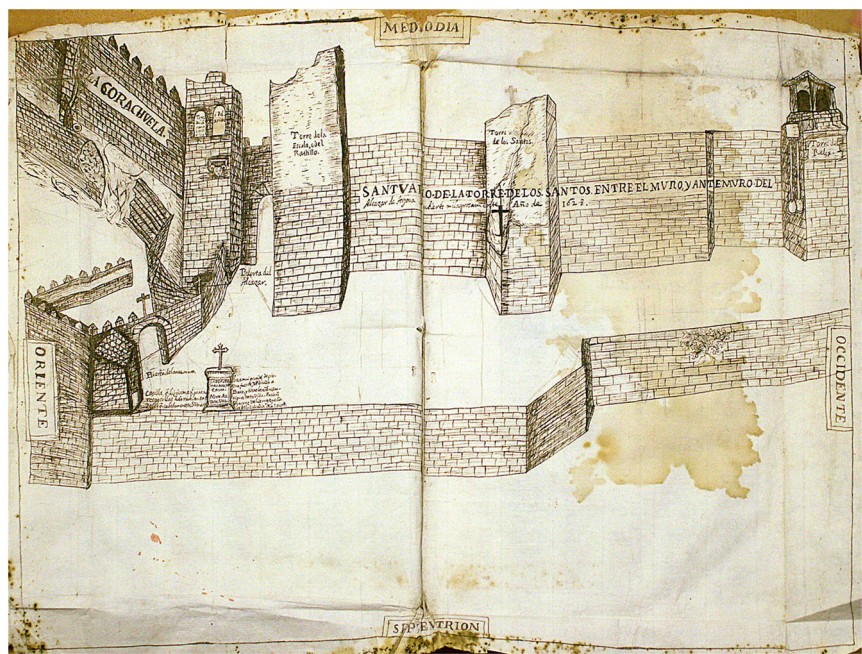

**Figure 14.** The sanctuary of the Tower of the Saints in Arjona drawn by Martín Ximena Jurado (1642, plan 5). Centro Documental y Biblioteca del Instituto de Estudios Giennenses, Diputación Provincial de Jaén, sign. MAP-B 201-5.

In chapter 29 of his history of the town of Arjona, he drew several idealised scenes of the martyrdom of the Roman soldiers Bonoso and Maximianus in the town of Arjona (Ximena Jurado 1643, pp. 116–18). He depicted three graphic sequences in the manner of

comic vignettes. In these images, he also drew the walls, towers, and gates of the Alcázar following a similar criteria of representation to the previous ones (Figure 15). He also drew a frontal territorial view of the section of the Guadalquivir Valley as it passed through the eastern part of the Jaén countryside (Figure 16), the place where he came from, placing his home town—Villanueva—at the centre with the royal road called "el Arrecife" (from Arabic *al-raṣīf*) going through it. The town of Arjona was placed in the lower left-hand corner of this drawing (Ximena Jurado 1643, p. 121).

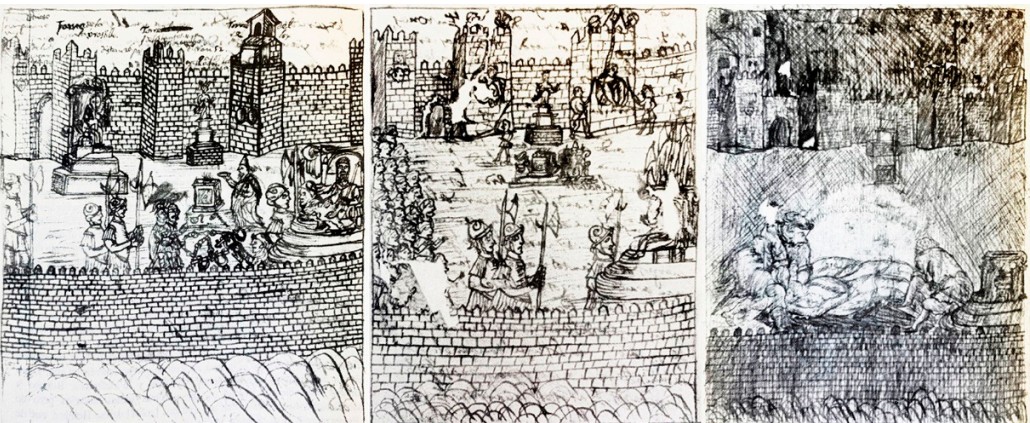

**Figure 15.** The *Alcázar* of the town of Arjona in three scenes drawn by Martín Ximena Jurado of the martyrdom of the Roman soldiers Bonoso and Maximianus (Ximena Jurado 1643, pp. 116–18. Arjona Municipal Archive).

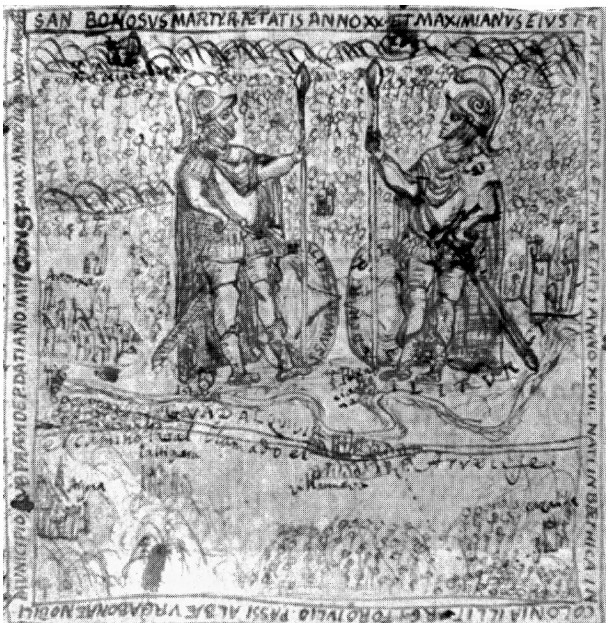

**Figure 16.** A territorial view of the section of the Guadalquivir Valley as it passed through the eastern part of the Jaén countryside drawn by Martín Ximena Jurado (1643, p. 121. Arjona Municipal Archive).

### 2.2. Walls and Towers of Eight Rural Castles and Small Fortified Enclosures

These drawings are usually marked with the cardinal points and measurements in paces. Their overall dimensions are indicated in the accompanying text.

The castle of Mengíbar (Figure 17). Ximena Jurado depicted a very simplified sketch with a dimension in paces in fol. 41 of *Ms. 1180 B.N.* (Eslava Galán 1999, pp. 195–98).

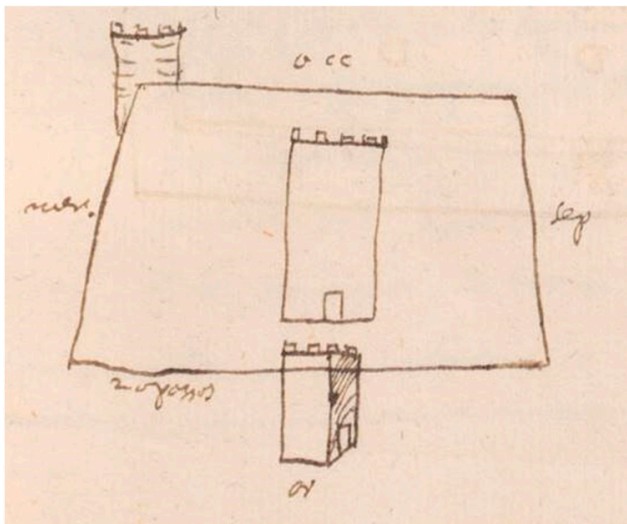

**Figure 17.** Sketch of the castle of Mengíbar drawn by Martín Ximena Jurado. *Ms. 1180 B.N.*, fol. 41.

The castle of Tobaruela, in the municipal district of Linares (Figure 18). He drew a small sketch in fol. 93 of *Ms. 1180 B.N.* (Eslava Galán 1999, pp. 265–66). On the other hand, fol. 170, intended for this location, is empty.

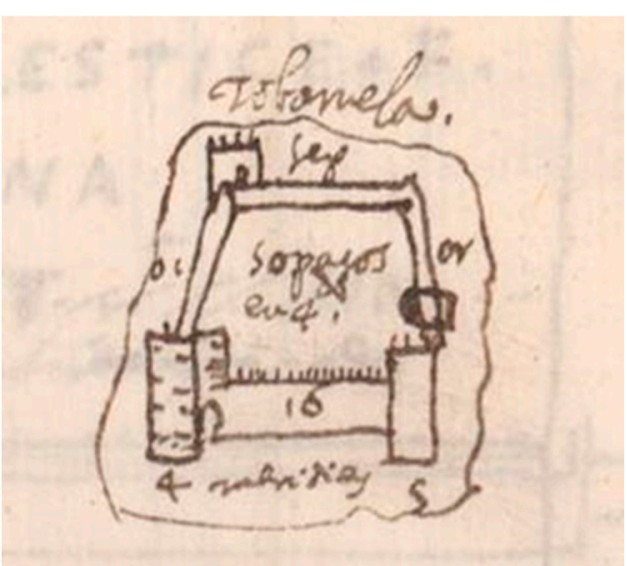

**Figure 18.** Sketch of the castle of Tobaruela (Linares) drawn by Martín Ximena Jurado. *Ms. 1180 B.N.*, fol. 93.

The castle of Marmolejo (Figure 19). Drawn in fol. 136 of the *Ms. 1180 B.N.* (Eslava Galán 1988, pp. 101–2, 111; 1999, p. 109).

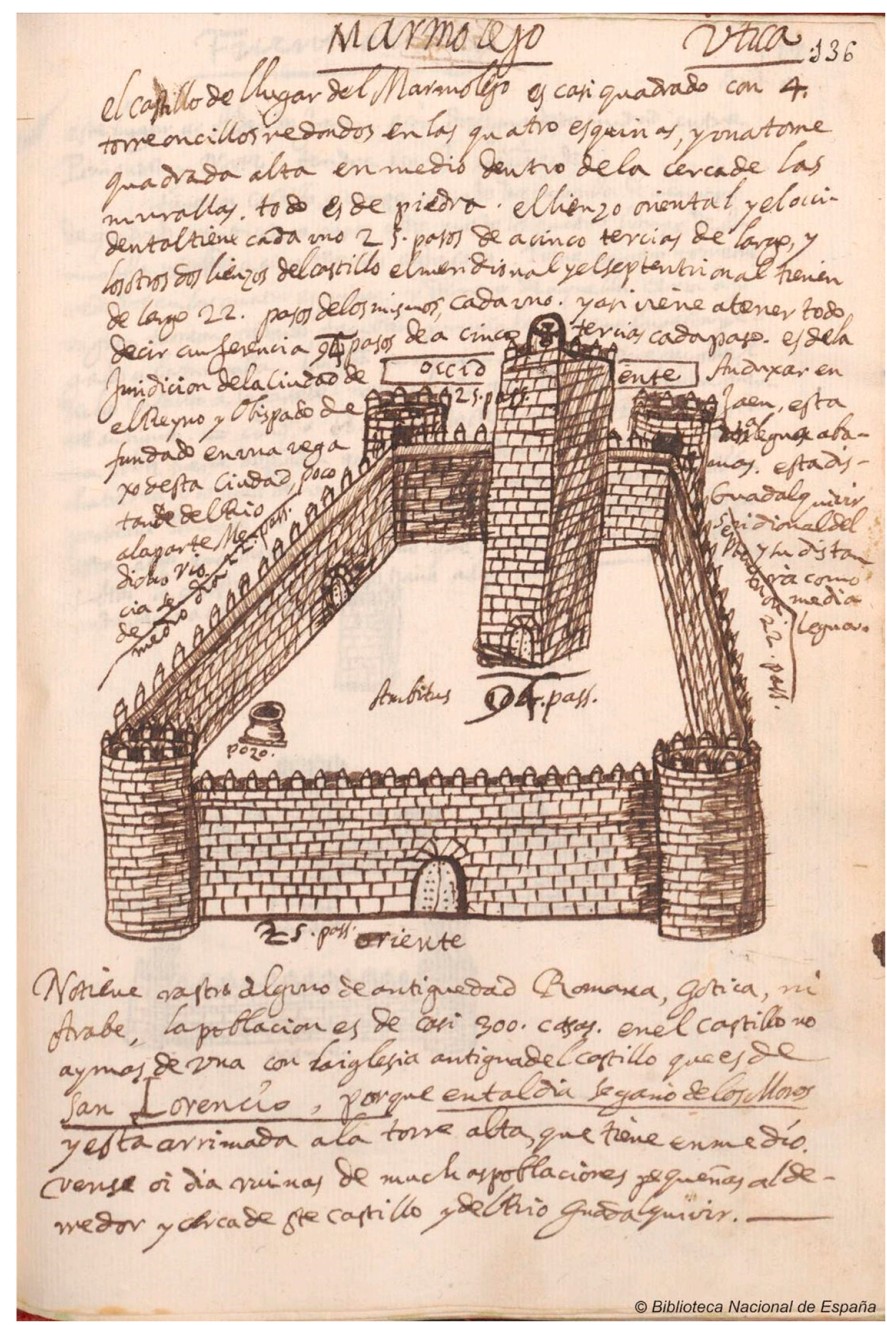

**Figure 19.** Sketch of the castle of Marmolejo drawn by Martín Ximena Jurado. *Ms. 1180 B.N.*, fol. 136.

The castle of Fuente del Rey (Figure 20). Sketched in fol. 137 of *Ms. 1180 B.N.* (Eslava Galán 1988, pp. 100–1, 110; 1999, pp. 107–9).

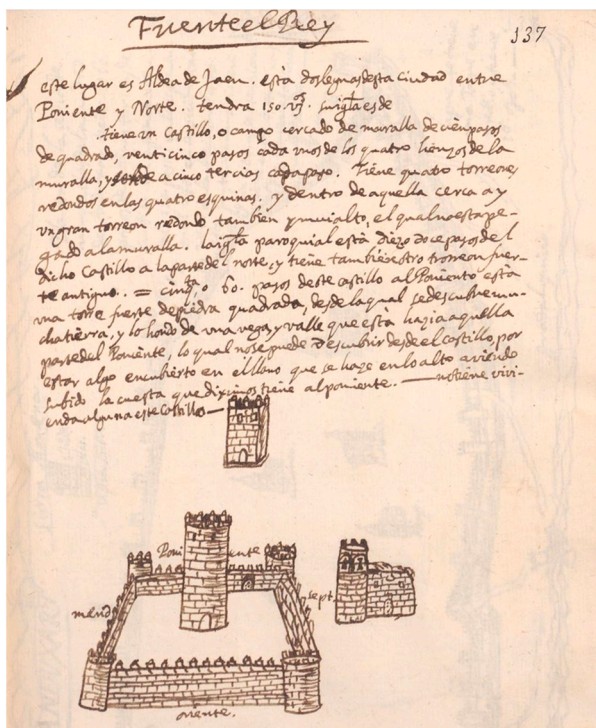

**Figure 20.** Sketch of the castle of Fuente del Rey drawn by Martín Ximena Jurado. *Ms. 1180 B.N.*, fol. 137.

The castle of Aragonesa or Bretaña, in the municipal district of Marmolejo (Figure 21). Depicted in fol. 139 of *Ms. 1180 B.N.* (Eslava Galán 1988, pp. 97–99, 104–7; 1999, pp. 103–5).

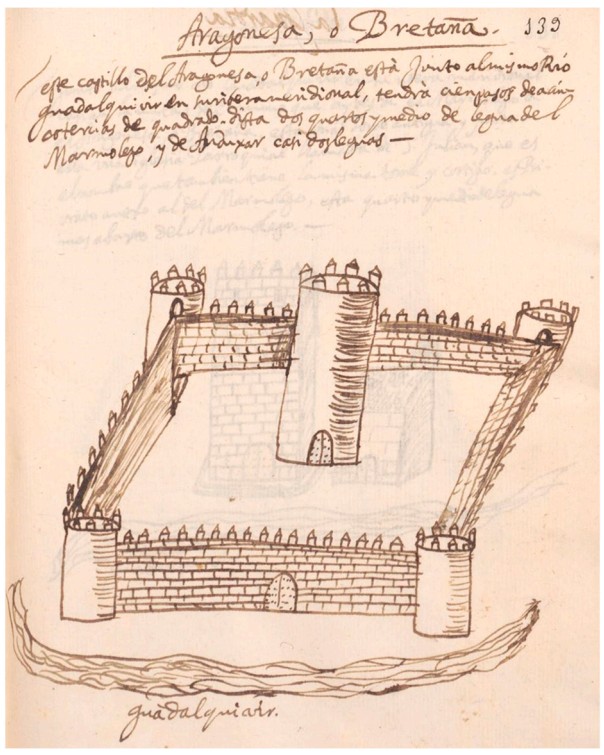

**Figure 21.** Sketch of the castle of Aragonesa or Bretaña (Marmolejo) drawn by Martín Ximena Jurado. *Ms. 1180 B.N.*, fol. 139.

The castle of El Aldehuela, in the municipal district of Andújar (Figure 22). Drawn in fol. 141 of *Ms. 1180 B.N.* (Eslava Galán 1988, pp. 100, 108; 1999, pp. 105–6).

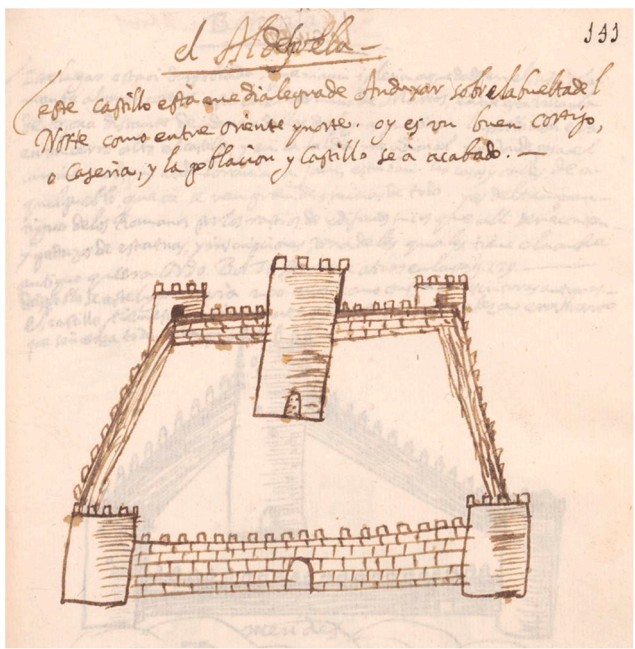

**Figure 22.** Sketch of the castle of El Aldehuela (Andújar) drawn by Martín Ximena Jurado. *Ms. 1180 B.N.*, fol. 141.

The castle of Benzalá, in the municipal district of Torredonjimeno (Figure 23). Sketched in fol. 142 of *Ms. 1180 B.N.* (Eslava Galán 1999, pp. 222–23).

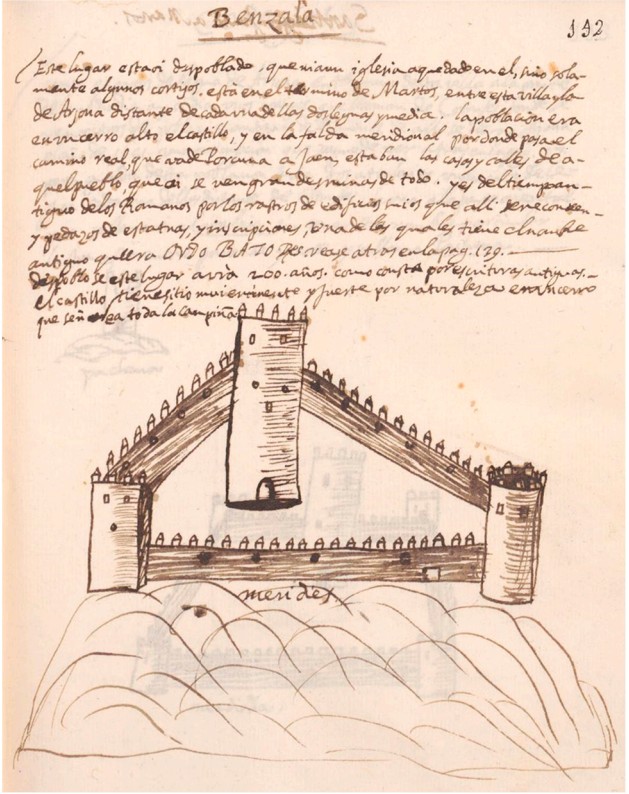

**Figure 23.** Sketch of the castle of Benzalá (Torredonjimeno) drawn by Martín Ximena Jurado. *Ms. 1180 B.N.*, fol. 142.

The castle of Cotrufes, in the municipal district of Arjona, plus two isolated towers, Atalaya (Watchtower) and Pachena (Figure 24). Depicted in fol. 143 of *Ms 1180 B.N.* (Eslava Galán 1988, pp. 100, 109; 1999, pp. 106–7).

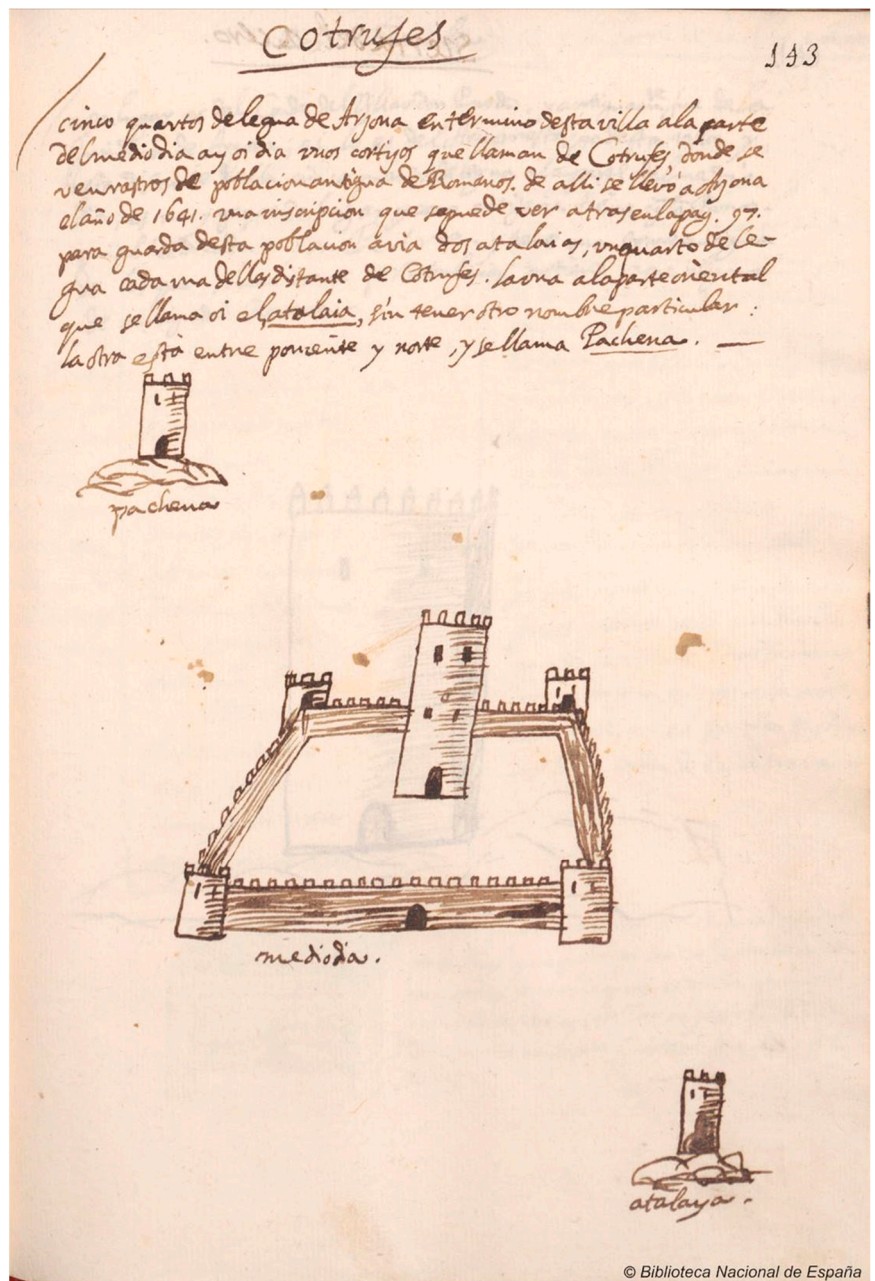

**Figure 24.** Sketch of the castle of Cotrufes and the towers of Pachena and Atalaya (Arjona) drawn by Martín Ximena Jurado. *Ms. 1180 B.N.*, fol. 143.

Among them are two small simple sketches, introduced in the chapters dedicated to antiquities located in the town of Mengíbar (fols. 40–41) and in the castle of Tobaruela (fols. 92–94). In the latter, Ximena Jurado located up to six reused stone elements with Latin inscriptions that were inserted in the doorway, courtyard walls, staircase, and towers.

In the remaining six cases, detailed drawings were made on the pages dedicated to the information on said locations. Two of them have since become municipalities and eliminated most of the remains of the fortifications during the 19th–20th centuries, as in the case of Marmolejo and Fuente del Rey. In two other cases, El Aldehuela (in the municipality of Andújar) and Cotrufes (in the municipality of Arjona), they were converted

into farmhouses, which transformed the castle that stood there beyond recognition. Another of the places is nowadays an archaeological site where part of the elements represented can still be seen such as the remains of the three towers represented in the castle of Benzalá (in the municipality of Torredonjimeno). The castle of Aragonesa (in the municipality of Marmolejo) is the best preserved, as the farmhouse that was attached to it in the east left a large part of the medieval fortification untouched. Furthermore, both the main tower and its circuit of walls with three other corner towers can still be seen.

These last six cases had similar dimensions and were located on strategic communication routes in the western part of the Kingdom of Jaén. Moreover, they were depicted with common formal characteristics such as a large keep within the quadrangular walled and crenellated enclosure (except in the case of Benzalá, where it is triangular), in which there were cylindrical turrets at the corners (except in the south-western corner of Benzalá). The main interior towers were also drawn as cylindrical, except in Marmolejo, whereas those preserved in Aragonesa and Benzalá are prismatic (Eslava Galán 1988, pp. 97–113).

### *2.3. Isolated Towers*

Three sketches were drawn of several towers with ancient inscriptions on their walls or were located next to religious buildings.

### 2.3.1. Tower of Cazadilla (Figure 25)

In fol. 36v of *Ms. 1180 B.N.*, this tower is drawn in a cylindrical shape and with battlements, with a building attached to the right of the inscription in Punic characters (Mozas Moreno 2018, p. 155), which was the main object of this representation. The attached body seems to be drawn in perspective, with battlements and a door with a semicircular arch on its lower part. He also drew this tower, but entirely isolated, with a gate and with ashlar coating, in his history of the town of Arjona and provided further documentation about the inscription:

> 'From the Hebrews who inhabited this land there is an inscription on an ancient tower in the town of Cazalilla, at the top of it, in one of the two corners that it has, because the plan of the tower is a semicircle and it lies two and a half leagues from Arjona to the east and half a league from the Guadalquivir river on its southern bank.' (Ximena Jurado 1643, p. 148)

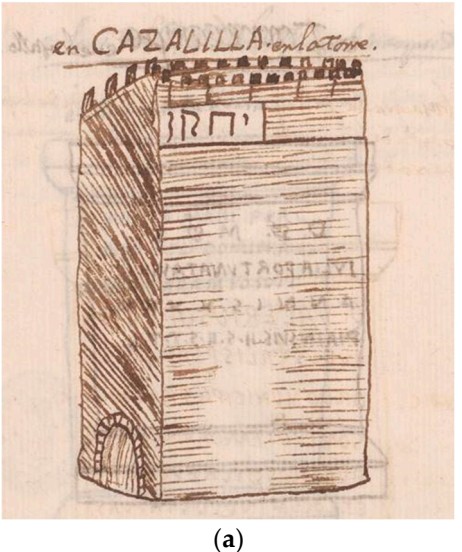
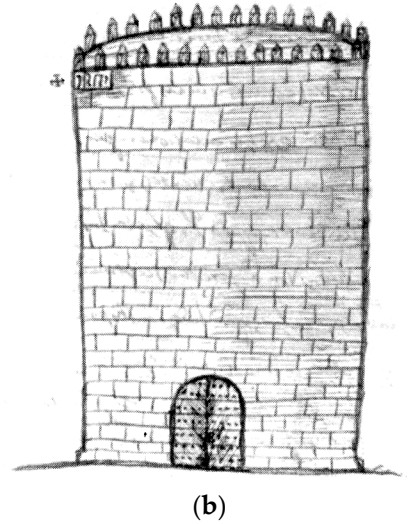

(**a**)      (**b**)

**Figure 25.** Sketch of the tower of Cazadilla drawn by Martín Ximena Jurado. (**a**) *Ms. 1180 B.N.*, fol. 36v; (**b**) and also published in his history of the town of Arjona (Ximena Jurado 1643, p. 148. Arjona Municipal Archive).

Today, it is integrated into the Church of Santa María Magdalena de Cazadilla. It is a solid semicircular masonry wall that was reused from a large cylindrical tower.

2.3.2. Tower of San Julián, in the municipal district of Marmolejo (Figure 26)

Fol. 140 of *Ms. 1180 B.N.* is dedicated to this site on the banks of the Guadalquivir, with the following explanatory text:

'Next to the same river Guadalquivir, on its southern bank, almost halfway along the road from Marmolejo to Aragonesa or Bretaña, there is an old tower, and next to it a parish church called San Julián, which is the name also given to the same tower and farmhouse; it is a priory annexed to that of Marmolejo; it is a quarter and a half of a league below Marmolejo'.

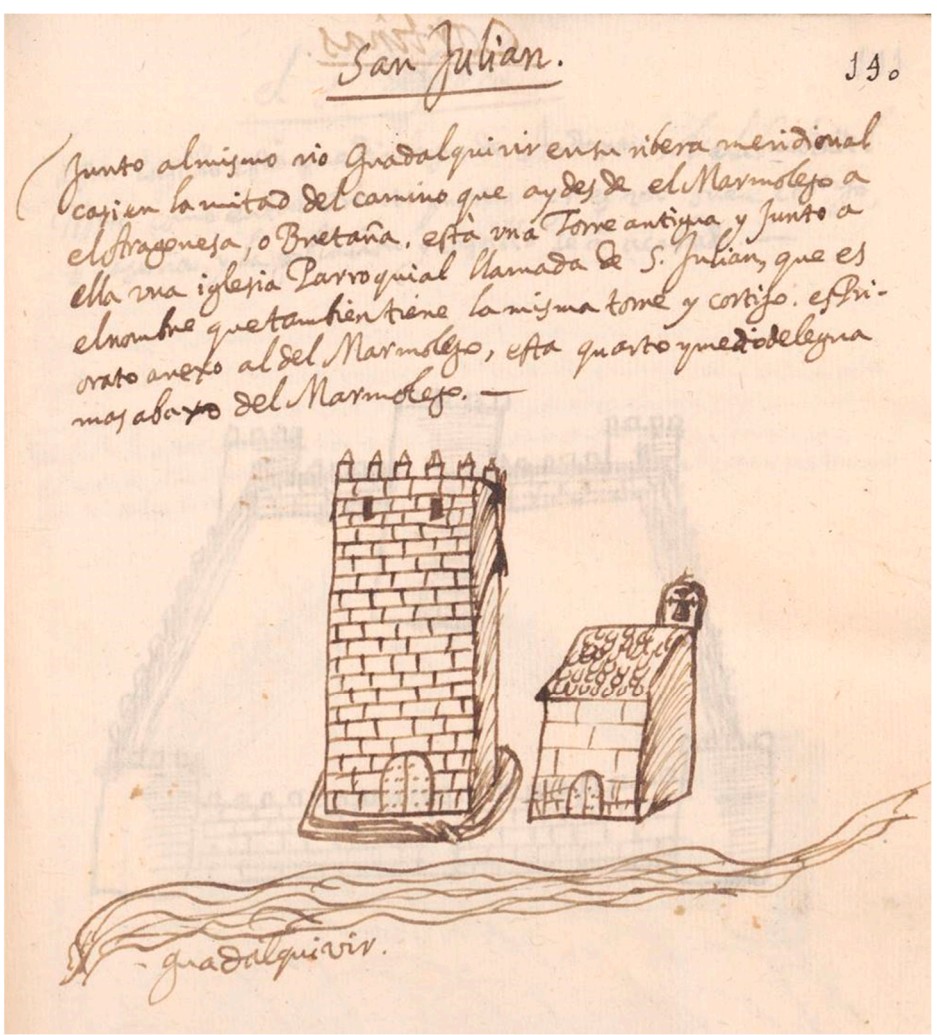

**Figure 26.** Sketch of the tower of San Julián (Marmolejo) drawn by Martín Ximena Jurado. *Ms. 1180 B.N.*, fol. 140.

Below, the author produced a perspective drawing of the crenellated prismatic tower, raised on a plinth with an ashlar quartering, a low door, and two arrow slits on its elevation facing the river, to the right of which a religious building with a chapel on its single-pitched roof has also been depicted.

No remains of the fortification of San Julián have been preserved (Muñoz López 1993). The site lies at a large meander in the Guadalquivir with two small river islands, now occupied by a hamlet located about 5 km south-west of Marmolejo, built in the mid-20th

century. The ancient Roman road from Cordoba that followed the Guadalquivir, known in the Middle Ages as the "Arrecife", used to run through this area.

### 2.3.3. Tower of Escañuela, in the municipal district of Arjona (Figure 27)

Represented in fol. 144 of *Ms. 1180 B.N.*, together with the following text:

'This place belongs to the Count of Villa don Pardo, and formerly belonged to the jurisdiction of Arjona; today it belongs to the archpriesthood of this town, and lies at a distance of five quarters of a league to the south from it [...], and has an ancient tower of those that used to be seen as watchtowers; it has no trace of Roman antiquity.'

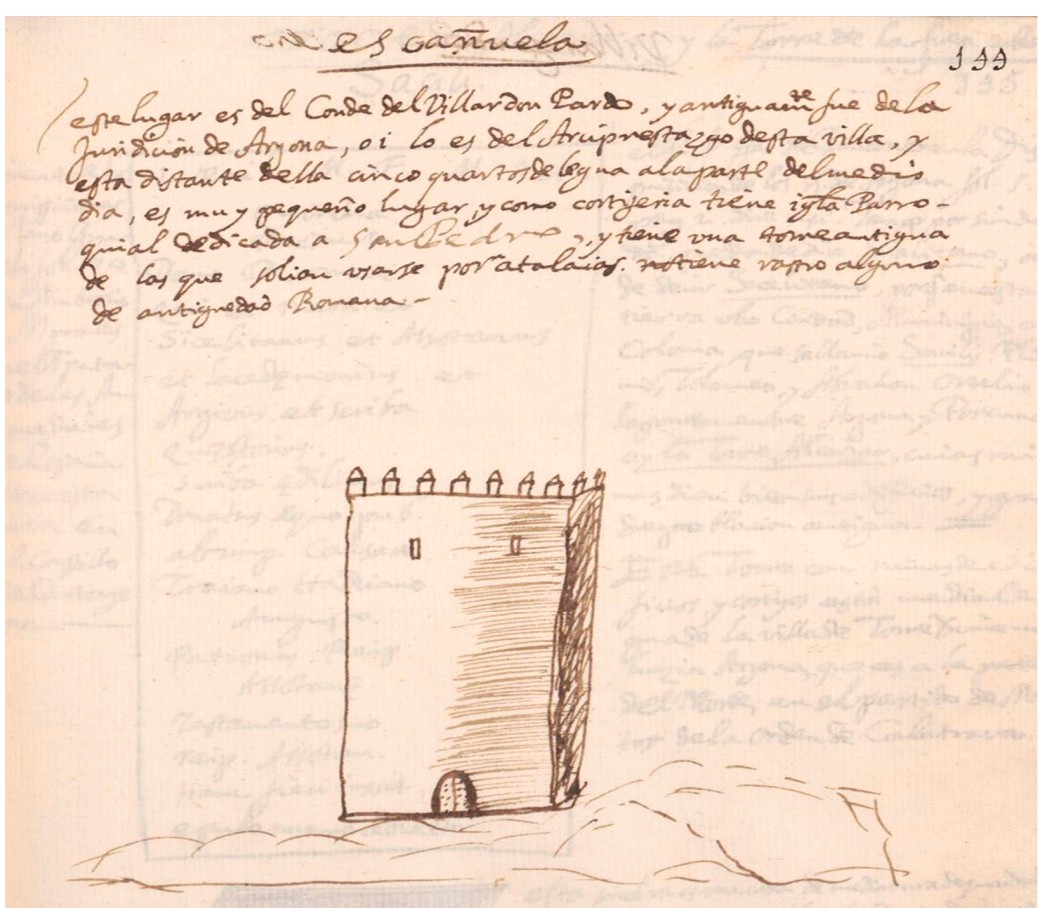

**Figure 27.** Sketch of the tower of Escañuela (Arjona) drawn by Martín Ximena Jurado. *Ms. 1180 B.N.*, fol. 144.

The design of the tower is very similar to that of San Julián, except that it does not have ashlars, but a horizontal striping on the right side of the main elevation, and the tower is located on what appears to be a rocky relief.

This tower is already mentioned in the donation of Escañuela as the lordship of Pedro Ruiz de Torres in 1385. It was requested again in 1394 in the lordship of Villardompardo, since it had been lost when the site was burnt and destroyed in some of the Nasrid raids carried out at the end of the 14th century.

The territory of Arjona at this time corresponded approximately to the present-day municipal district of Arjona, Higuera de Arjona, Arjonilla, and Escañuela. Within this area, there were several small villages such as Cotrufes, Pachena, Herrerías, Corbús, Hadón, Escañuela, Arjonilla, and Almoraide. In all of these places, there would have been small fortifications that acted both as watchtowers and military outposts of Arjona and also temporarily defended the surrounding population in the case of sudden danger.

In his manuscript, Ximena Jurado also presents—although without drawings—several towers that had antique stones with inscriptions enclosed in its walls. Among them, he wrote about the old remains found in the Tower of Alcázar of Bailén (fol. 90v), in the ruins of Tower of Fuencubierta (fol. 145), or in Tower of Santa Potenciana in Villanueva de Reina (fols. 252–253). He also described the remains of ancient buildings next to the aforementioned towers, the ruins of "Turbula" next to the Tower of Tobaria (fol. 89v), or the Tower of Alcázar in the district of Torredonjimeno (fol. 145).

## 3. Discussion

Had Ximena Jurado completed this chorographic work, it could have emulated, on a regional scale, the great work *Descripción de España y de las costas y puertos de sus reinos* ('Description of Spain and the coasts and ports of its kingdoms'), initiated a few years earlier by the Crown's Head Cosmographer, Juan Bautista Lavanha (1555–1624). This work was commissioned by Philip IV of Spain (r. 1621–1665), and in 1622, Pedro Texeira Albernas (1595–1662) joined the project, finishing it alone in 1634. It included the mapping of the coasts of the Iberian Peninsula including the fortifications. Its author was a member of an illustrious family of Portuguese cartographers who had settled in Spain in the early 17th century. His father, Luis Texeira, was Head Cosmographer of Portugal and had already made a series of charts that influenced the work of his son.

The result was a compendium of views and maps that is considered a masterpiece of 17th-century cartography, combining high artistic value with a degree of scientific accuracy for its time. It was intended as a complete description of the coasts of Spain and Portugal when the two countries formed a single kingdom between 1580 and 1640. Texeira's atlas included the main Iberian ports and the most important coastal cities, with notes on antiquities and history painted in tempera. It consisted of two parts, a literary description with information on the geography, history, and population of the territory and 116 cartographic cards (11 of Guipúzcoa; five of Vizcaya; five of Castile; nine of León; 19 of Galicia; 21 of Portugal; 16 of Andalusia; two of Murcia; five of Valencia, and seven of Catalonia), representing places and place names at different scales (Pereda Espeso and Marías Franco 2004, pp. 129–57).

The images in this atlas have in common with those of Ximena Jurado the fact that they were always pseudo-perspective aerial representations; Texeira's work was taken from a point high above the sea, imagined in the author's mind from the observations he made, sometimes from a boat, or from the information and descriptions he received. The chosen system of representation must have been selected because it made the cartographic information easier and more visual to read. Hence, the drawing criteria are not absolute in their realism, but are designed to be quick and easy to interpret, despite the apparent naivety of their depiction.

Also contemporary to Ximena Jurado's work is *Livro das Plantas de todas as fortalezas, cidades e povoações do Estado da Índia Oriental* ('Book of plans of all the fortresses, towns and settlements in the state of the East Indies'). In 1632, Philip IV of Spain ordered the Viceroy of the State of India, Fernando de Noronha, V Count of Linhares, to survey the fortresses under his jurisdiction in East Africa, the Arabian Peninsula, and the Far East (Garcia 2009). The latter, in turn, in 1633, entrusted the task to the official chronicler of the State of India and Chief of the Guard of the Royal Archives of Goa, António Bocarro. The result was the aforementioned atlas, created in two originals and sent to the court in Lisbon in early 1635. Included in the text were fifty-two maps of fortresses and cities. Although Bocarro did not name the author of the drawings, it is now known that they were made by Pedro Barreto de Resende, secretary to the Count of Linhares, as he states at the beginning of the codex *Descriptions of the Fortresses of the East Indies,* written by himself. Other versions of the manuscript include the *Livro do Estado da Índia Oriental* (ca. 1636) by Pedro Barreto de Resende himself. The work is presented in the form of an album with 48 historically and iconographically important engravings. These are attractive for their polychromy, although they lack essential elements such as scale and cardinal orientation, a fact that António

Bocarro himself regretted. The representations are carried out without much carefulness and rigour, plant symbols are abundant, and the houses and walls of the fortresses are depicted in large proportions.

These representations from the first half of the 17th century had magnificent precedents in the Iberian Peninsula in other chorographies of fortresses made in the previous century.

The first of these (de Armas [1510] 1997) would have been the *Livro das fortalezas situadas no extremo de Portugal e Castela por Duarte de Armas, escudeiro da casa do rei D. Manuel I* ('Book of the fortresses located in the extreme of Portugal and Castile by Duarte de Armas, squire in the house of King Manuel I').

The Lisbon-born Duarte de Armas (1465–ca. 1516), a notary and scrivener at the Livraria Régia and Torre do Tombo, quickly stood out for his drawing skills. He was commissioned by King Manuel I of Portugal (r. 1495–1521) to draw up the state of the fortifications on the border with Castile based on direct observation. He conducted this between 1509 and 1510 and set it down in a codex based on plan drawings and panoramic views depicted in pseudo-perspective with measurements, cartographic annotations, and explanatory texts of 56 fortresses, from Castro Marim to Caminha, which also included 110 plans relating to 55 of these settlements on the border between Portugal and Spain. This is a truly sui generis book, with few international parallels, and is now a valuable source for the study of cartography and ancient military architecture in Portugal (Nunes da Silva 2019, pp. 190–223).

At the same time that Ximena Jurado was writing the *Ms. 1180 B.N.*, another Portuguese author, Brás Pereira, copied 55 of Duarte de Armas' drawings and watercoloured them, publishing them in 1642 in the book entitled *Fronteira de Portugal fortificada pellos reys deste Reyno* ('Border of Portugal fortified by the kings of this kingdom').

Other notable cases include the city of Granada, where a rich iconography has been preserved from around 1500. It was captured in the fresco of the Battle of the Higueruela in the Hall of Battles at El Escorial and in the panel Virgin and Child by the Flemish artist Petrus Christus II, in which the walls, towers, and other defensive elements of what had been the capital of the Nasrid kingdom were depicted (Orihuela Uzal 2001, pp. 103–34).

The last quarter of the 16th century is noteworthy for the well-known drawings of the city walls by the Flemish painters Anton van den Wyngaerde and Joris Hoefnagel.

The Antwerp-born Anton van den Wyngaerde (1525–1571) was trained as an illustrator in the Netherlands and travelled through Italy between 1552 and 1553. He was hired by Philip II of Spain in 1557 as an 'ordinary painter'. He settled with his family in Madrid in 1562 at the request of the king, devoting himself to the chorographic description of Spanish towns, villages and cities until his death in this capital in 1571. To this end, he began to travel around the Iberian Peninsula from 1561 onwards, drawing a collection of sixty-two views of Spanish cities in pen and ink, many coloured with watercolours. This graphic legacy that he left behind consisted of drawings from life and imagined aerial panoramas in which meticulous detail and topographical descriptions are paramount. Years after his death, his works reached the workshop of an engraver, as demonstrated by the fact that some of them have a grid superimposed to enable their reproduction with a burin.

He was not the only painter of cities, as Joris Hoefnagel (1542–ca. 1600), also an Antwerp-born painter, travelled around Spain between 1563 and 1567 and produced drawings for the seminal work *Civitates Orbis Terrrarum* by Braun-Hogenberg, publishers of the atlas of cities to which he was the main contributor. In total, he provided 63 views, 44 of which were of Spanish towns and 34 centred on Andalusia, together with enriching accounts of his own experiences. These drawings may have been influenced by those of Wyngaerde (Richard Kagan 2000). His pictorial conception, eminently scenographic, was less meticulous, with a tendency to over-dimension the natural surroundings, seeking to increase the spectacular effect of the urban landscapes, in contrast to the topographical precision of other artists. His views are compositions conceived on the basis of sketches drawn in situ, combining topographical criteria with the design of the urban landscape and pictorial reproductions of different facets of everyday life of undoubted anthropological

value. He introduced, preferably in the foreground of the perspectives, social, cultural, and occupational representations.

Hoefnagel lived in Andalusia for most of his stay on the Iberian Peninsula, allowing himself to be seduced by its charm and the still evident Muslim imprint, thus anticipating the romantic fervours of later centuries.

Rotogravure techniques, applied to printing by means of expertly chiselled plates to reproduce drawings, facilitated the rapid and massive dissemination of these prints, making them accessible to the working classes. They were soon used mainly to highlight the image of political power, becoming an effective propaganda tool.

In contrast, Ximena Jurado's graphic representations have been preserved as black pen drawings, as manuscript representations for his books. In his catalogue of the bishops of Jaén, some of his small-format drawings referring to antiquities were engraved and inserted in the text. This can be seen by comparing the drawings of the ashlar with a bull engraved in the stone of the tower next to the hermitage of Santa Potenciana (fols. 36 and 252 of *Ms. 1180 B.N.* and Ximena Jurado 1654, fol. 24). On the other hand, the plan of the "Descripción del Reyno y Obispado de Jaen", which he included in fol. 203 of *Ms. 1180 B.N.* to situate the locations that interested him in his study (Figure 1), may have served as the basis for the "Mapa del Reyno y Obispado de Jaén" that the engraver Gregorio Fosman y Medina (ca. 1635–1713) made to accompany the editions of the books by Bilches Pedraza in 1653 (*Santos y santuarios del Obispado de Jaén y Baeza*) and by Ximena Jurado in 1654 (*Catálogo de los obispos de las iglesias catedrales de Jaén y Anales eclesiásticos de este obispado*).

## 4. Materials and Methods

The analysis of the structure and content of the manuscript *Ms. 1180 B.N.* and of his other books allows us to understand what data Ximena Jurado was looking for. His main interest in antiquarian research was aimed at documenting the territory in Roman times, a period associated with the relics of the Christian martyrs as well as the pre-Roman culture of 'the first Spaniards', provided by material remains and literary sources, and, of course, trying to provide data and the remains of early Christianity in this part of the Iberian Peninsula.

This methodology, developed by authors such as Ambrosio de Morales[6] (1513–1591), consisted of observing antiquities in situ, trying to provide chronologies from the material remains. The aforementioned author listed a series of points to be taken into account in the historiographic study of the material remains observable on archaeological sites (Mozas Moreno 2018, p. 89). This methodology, which was applied by Ximena Jurado when compiling the data in *Ms. 1180 B.N.*, was systematised in the following points:

- Numismatic and epigraphic studies were carried out.
- Inspection of the depopulated areas and the location of abundant fragments of pottery on the surface.
- Drawings of fortified enclosures with notes of their measurements. Graphic annotations would also be made of drawings of statues and other material remains in situ or decontextualised, trying to locate the site of origin.
- Study of the texts of Greco-Latin historians and geographers. One of the most important pieces of information they could provide was the location and ranking that Ptolemy gave to the different cities.
- Use of ancient itineraries to study Roman roads at a regional level.
- Documentation of literary texts with quotations or drawings from different ancient or recent writers.
- Study of the toponymy of the places, focusing especially on that of Latin origin.
- Consultation of the history of the Councils of the Catholic Church and the martyrologies.
- Soliciting the advice of illustrious people and listening to the opinion of the natives and neighbours of a locality.

Together with another antiquarian, teacher, and historian from Baeza in the mid-17th century, Francisco de Rus Puerta (late 16th century–1649), who also made a great effort to

integrate the architectural and monumental remains within the topographical framework, Ximena Jurado was a follower of this trend. These two men left in their testimonies, in the form of literary works, tools to know how these antiquities had been investigated, especially in the cities where their presence was evident. In addition, they tried to identify the delocalised place names mentioned in the Greco-Latin sources, especially with regard to the location of the primitive episcopal seats, with the aim of elaborating an ecclesiastical history of the diocese of Jaén. However, his interest in this prerogative would make him move away from the more rigorous method of Ambrosio de Morales, or of Francisco de Torres, who was a radical advocate of an antiquarian methodology consisting of looking at the ground and describing the vestiges and archaeological sites. In contrast, the results and conclusions often reached by de Rus Puerta or Ximena Jurado were based fundamentally on the evidence provided by coins and inscriptions, which is why false chronicles can be found in some of the studies that make up *Ms. 1180 B.N.* However, in many other cases where he did not employ fantastic interpretations, the information he provided is genuine documentation (Mozas Moreno 2018, p. 87).

Thus, Ximena Jurado stood out for being a pioneer of the historical cartography of this Hispanic region. This can be seen in his maps of the kingdom and bishopric of Jaén (Figure 1) from 1641 to 1654, where he tried to represent both the settlements that were inhabited in his time and the depopulated and emblematic ancient places, their ruins, and fortifications. The data and drawings of castles, towers, and walled enclosures that he included in his texts show a special interest in the militarisation of cities and their territory. He mapped ruins and drew fortifications in his manuscript, accompanying them with various texts. He undertook exceptional work given the enormous wealth of the remains of cities and fortified sites in the Kingdom of Jaén (Mozas Moreno 2018, pp. 482–84).

## 5. Conclusions

The views of Iberian cities produced in the 16th and 17th centuries must be seen in the context of the splendour and popularity acquired by the chorographic genre in the period called the Spanish Golden Age (ca. 1492–ca. 1681). Thus, the images that accompanied the chorographic descriptions were a fundamental complement to these works.

In Spain, the genre developed in several directions at the same time. On the one hand, it was frequently used by cosmographers and geographers such as the Cordovan Hernando Colón y Enríquez de Arana (1488–1539) with his work *Descripción y Cosmografía de España*, a project initiated in 1517 as an itinerary in which he described 'all the particularities and memorable things'. On the other hand, the genre was often found in works of humanist writers such as the Sicilian Lucio Marineo Sículo (1444–1536) with his work in 1530, *De las cosas memorables de España*. Other authors can be highlighted such as Pedro de Medina (ca. 1493–1567) with the books *Libro de* Cosmografía (1538) and *Libro de las grandezas y cosas memorables de España* (1548), the latter in the form of a great topographical compendium with chorographic descriptions of several hundred Spanish and Portuguese municipalities, in order to present Philip II with the kingdoms he was to inherit.

During the reigns of Philip II, Philip III, and Philip IV of Spain, chorography was, together with painting, put at the service of the Crown to demonstrate the greatness of the Habsburg monarchy. The representation of cities in the midst of transformation, however, still showing elements of their Islamic past with citadels, walls, and towers, became an essential part of the humanist cultural environment of this century.

These authors were not the only painters of cities, but were part of a European trend that began in the Renaissance and continued into the 16th and 17th centuries. Remarkable examples of this chorographic genre in a period close to that of Ximena Jurado can be found from south to north on the continent. Among them is the rich collection of drawings with panoramic views of the towns and villages through which the Florentine architect and painter Pier Maria Baldi (1630–1686) travelled, accompanying Cosimo de Medici on his journey through Spain, Portugal, France, Belgium, and Holland between 1668 and 1669 (Neira Cruz 2004). Another outstanding case is the extensive work of 353 engraving

plates compiled by the Swedish military engineer Erik Jönsson Dahlbergh (1625–1703) and published in the compendium *Suecia Antiqua et Hodierna* (1660–1716) (Jonsson 1992).

All of these great draughtsmen and illustrators had specific training in the graphic arts, the fundamentals of drawing, and the most advanced perspectival techniques of the time, which gave them the tools to capture their views of cities in the most realistic, accurate, and artistic way, so well that they are today considered masterpieces of cartographic production. Many of them travelled widely in Europe and were trained on the Italian Peninsula, where perspectival knowledge was advanced.

By the first third of the 15th century, the Italian artistic avant-garde had managed to establish a few rules for the depiction of the drawing space. This trajectory can be traced back to the end of the 13th century when Giotto di Bondone (1266–1337) developed an early intuitive drawing style, solidly three-dimensional with the accurate depiction of perspectives with one and two vanishing points. He was followed by great masters of the 14th and 15th centuries such as Ambrogio Lorenzetti (1290–1348), Bernardo Daddi (ca. 1312–1348), Andrea di Bonaiuto (ca. 1346–1379), Filippo Brunelleschi (1377–1446), Donato di Niccolò di Betto Bardi—Donatello—(ca. 1386–1466), Paolo Ucello (1397–1475), Stefano di Giovanni—Il Sassetta—(1392–ca. 1451), Tommaso di Ser Giovanni di Simone—Masaccio—(1401–1428), Lorenzo Ghiberti (1378–1455), Antonio Pollaiuolo (1429–1498), Andrea del Verrochio (ca. 1435–1488), Domenico Ghirlandaio (1449–1494), Sandro Boticelli (1445–1505), Francesco Raibolini—Francesco Francia—(1447–1517), or Antonello da Messina (ca. 1430–1479).

Although without following a complete systematisation, these artists had already adopted the single vanishing point for parallels, as would be the case in Flanders shortly afterwards. In the north of the continent, outstanding figures emerged such as Jan Van Eyck (1395–1441), who is considered as one of the forerunners of perspectival methods. Later on, Petrus Christus (1410/1415–1475/1476) developed a unitary conception of space by simplifying the models of his predecessor, which enabled him to empirically discover the basic rule of linear perspective by constructing his compositions with a single vanishing point. Dirk Bouts (1415–1475) also made use of this principle in some of his works.

It is also worth noting the development of the representation of architectural spaces and volumes from the latter part of the 13th century in the genre known as the Book of Hours or *Horarium.* These manuscripts are characterised by their extensive illumination and are a source of iconography of medieval Christianity. Architectural volumes resolved by means of conical and axonometric perspectives often form part of the composition. In some of these works, the powerful presence of castles, towers, and walled cities sometimes stands out in the background of the images.

Against this background, the standard of graphic representation that spread began to make use of conical perspective. Its origin cannot be established in ancient Hellenistic and Roman paintings, nor in the studies of Euclidean optics—rather, it started to develop in the Middle Ages. It came from a set of practical methods elaborated from the 13th century by European painters, who gradually began to adopt the convergence of parallels to a centred vanishing point of the composition, and even to two vanishing points in some elements of the scene (Grayson 1964; Arévalo Rodríguez 2003; Panofsky 2003; Neira Cruz 2004; Gentil Baldrich 2011; Ramón-Laca Menéndez de Luarca 2020).

Leon Battista Alberti (1404–1472) is considered to be the author of the first treatise written in Latin on conic perspective, found in his work *De pictura* (Rome, ca. 1435). It was the first of a trilogy of treatises, together with *De re aedificatoria* (ca. 1450) and *De statua* (ca. 1462), and was widely disseminated within the humanities. In this work, Alberti asserted: 'No one of understanding will deny that no painted thing can be seen to resemble reality unless a certain reason is employed'. The theory of graphic representation in perspective developed enormously with works such as those by Leonardo da Vinci (1452–1519), who, although he did not publish any specific work on the subject, in fol. 42r of his manuscript A, sketches the construction of a method of perspective. Also noteworthy are the treatises by Italians such as Sebastiano Piero della Francesca (*De Prospettiva Pigendi*, ca. 1475), Ponponio Guarico (*De sculptura: … De perspectiva*, 1504), Fra Luca Pacioli (*De*

*divina proportione*, 1509), Sebastiano Serlio (*Il secondo libro d'Architettura*, 1545, dedicated to perspective and scenography), Monsignor Danielle Barbaro (*La Prattica della Perspettiva*, 1569), the also cleric Egnatio Danti (*Le due regole della prospettiva prattica*, 1583, with the compilation of writings by M. Jacopo Barozzi da Vignola), or Lorenzo Sirigatti (*La pratica di prospettiva*, 1596).

In northern and central Europe, perspectival codes had become extraordinarily widespread, and had acquired a greater normative rigour to Italian models. Albert Dürer published *Underweysung der messung* in Nuremberg in 1525, which influenced many other German authors, with geometry being more important than perspective. Dürer was keen to depict perspective machines in his work, two in the 1525 edition, which increased to four in the 1538 edition. Other treatises that would have been influenced by him were Hieronymus Rodler (*Underweysung der kunts des Messens dem Zirckel Richtscheidt oder Linial*, 1531), Ulrich Kern (*Eyn new kuntsliche wolgegründt Visierbuch*, 1531), Walther Hermann Ryff (*Der neuen Perspectiva*, 1547), Heinrich Lautensack (*Des Circkels unnd Richtscheyts, auch der Perspectiva und Proportion der Menschen und Rosse, kurtze, doch gründtliche underweisung deß rechten gebrauchs*, 1564), Lorenz Stoer (*Geometria et perspectiva*, 1567), Wenzel Jamnitzer (*Perspectiva corporum regularium*, 1568), or Hans Lencker (*Perspectiva*, 1571).

From the beginning of the 16th century, France also saw a stream of treatises emerge, which began with the publication in 1505 of *De artificialis perspectiva*, by the canon of Toul (France), Jean Pélerin "le Viateur". It was written in Latin and French to 'proceed in representing and artificially figuring things seen or conceived'. In this work, Pélerin developed the rule of "tiers point" or distance points to represent oblique perspectives with two vanishing points on the horizon line. In his work, the drawings are more important than the text, taking from geometry only those lines that are necessary for the representation. Following this theory of perspective, to which Italy and northern Europe remained practically unfamiliar for much of that century, other Frenchmen also published treatises such as Jean Cousin (*Livre de Perspective*, 1560) and Jacques Androuet du Cerceau (*Leçons de perspective positive*, together with *Les plus excellents bastiments de France*, 1576). In them, the conic language began to transgress. In some illustrations, an axonometric pre-language appeared, representing parallel straight lines without a vanishing point. That would later give rise to the "military perspective", with the possibility of measuring in visual representations, given the geometric complexity of the layout of the conic. In military engineering, the design and construction of fortifications required them to be rationally measured, which gave rise to dimensionally controllable perspective systems. Another cleric, Father De Breuil, also referred to military perspective in his work *Perspective prattique par un religieux de la compagnie de Jesus* (1639).

This treatise was published in the same year in which Ximena Jurado may have begun to write *Ms. 1180 B.N.* However, despite this vast tradition that preceded him by more than two centuries, he does not seem to have been familiar with these avant-garde movements, nor the gradual evolution of perspective since the 13th century in paintings and reliefs with Christian themes. In addition, among these treatises, there have been a number of publications by religious authors. However, Ximena Jurado's depictions do not systematically reflect any of the perspectival theories developed up to that time, currents to which Spain had not remained on the fringes. In this respect, one might point to Cristóbal de Rojas's treatise *Teorica y practica de fortificacion, conforme las medidas y defensas destos tiempos: repartida en tres partes* (1598), which appeared in Madrid several decades before Ximena Jurado's drawings of castles, towers, and walled cities.

In contrast, he is shown to have been self-taught in graphic representation and had possibly never learned drawing techniques, given his limitations in accurately capturing the reality he observed. The general characteristics of his depictions are as follows:

- He was unaware of the fundamentals of the laws of perspective that had been widespread in the Renaissance. As a result, his representation system was very elementary, but coherent. In his views and plans, he intuitively introduced mixed conic, axonometric (mainly pseudo-military perspective), and dihedral techniques,

but without consciously defining any of them. In this way, the drawings are often incorrectly executed, with the walls being placed in a position that is impossible from the observer's main point of view.

- In *Ms. 1180 B.N.*, the sketches present different degrees of elaboration depending on the time invested in their creation, made with one or two inks (black and brown, sometimes with parts retouched in one or the other colour afterwards), without using gouaches to generate chiaroscuro. The drawings he published in his account of the discoveries in Arjona (Ximena Jurado 1642, plans 1 to 5) were more elaborate, using more resources of colour differentiation and even gouaches (Figures 12 and 13). In contrast to the drawings of the towers and walls where the relics of Arjona were found, the medieval fortifications that he mapped were generally not drawn as ruins, as they were in the mid-17th century. Instead, he depicted their most characteristic elements of construction, representing the idealised hypothesis of the state of these constructions at the time of the Castilian conquest of this territory in the first half of the 13th century and from then onward.

- Ximena Jurado was always interested in showing the orientation of the views in relation to the four cardinal points in written form, but without using the compass rose. Only in a few drawings, two of Arjona and another of the sites and villages of the bishopric of Jaén, were graphic scales introduced; in the latter case with a 'scale of leagues of four miles, which are the leagues common in Spain' and a 'scale of degrees and minutes' represented in the grid in which the map was framed.

- Some of the details are quite precise, but others are less accurate and furthermore inconsistent with reality, with different layouts of the depicted elements (Figures 4 and 25). Not all of the drawings that were in his own handwriting were made at the site or in front of the original—in certain cases, he would have drawn them from the descriptions or sketches from third parties.

- The proportions between the different elements introduced in his drawings are not realistic, but may be atrophied or magnified depending on the relevance he wanted to give them.

However, despite all of these technical flaws and inaccuracies in the degree of fidelity of his drawings, the freshness with which Ximena Jurado made them, without major corrections to the fit as well as the attractive system of representation, that was easily understandable, make them a very valuable, direct, and, in many cases, verifiable piece of information. In addition, he provided documentation on numismatics, antiquities, buildings, and urban environments that have been partially or totally transformed since then. They are therefore an important key to understanding the organisation and design of the Andalusi defensive systems established in these locations, which have come down to us much more deteriorated and in smaller numbers than they were at the time when Ximena Jurado drew them.

**Funding:** The research was completed within the frame of the R&D project led by Luis José García-Pulido 'Graphic documentation of the medieval castles preserved in Andalusia. Updated knowledge and spread of architectural heritage (ALCAZABA)' (reference UMA18-FEDERJA-257), financed by the Operative Program FEDER Andalusia Operational Programme 2014–2020, University of Malaga, call of 2018. Publication costs were funded by CSIC (Consejo Superior de Investigaciones Científicas).

**Data Availability Statement:** The research data can be found in Ximena Jurado's preserved works in the following libraries and archives: Biblioteca Nacional de España (*Ms. 1180 B.N.*, original available online, and with a monography published by Mozas Moreno 2018), Centro Documental y Biblioteca del Instituto de Estudios Giennenses, Diputación Provincial de Jaén, sign. A-Z6, A-Y6, Z4 (Ximena Jurado 1642), Archivo Municipal de Arjona (Ximena Jurado 1643). *Catálogo de los obispos de las iglesias catedrales de Jaén y anales eclesiásticos de este obispado* (Ximena Jurado 1654) has a facsimile edition (Rodríguez Molina and Osorio López 1991). The printed 17th-century version can be found in the library of the University of Granada, among others.

**Conflicts of Interest:** The author declares no conflict of interest.

## Notes

<sup>1</sup>  It was given this name in 1958 by the Biblioteca Nacional de España (Spanish National Library) when it was described in volume IV of the general inventory of manuscripts and numbered as *Ms. 1180 B.N.* It contains close to 350 folios, with most of them measuring 208 × 100 mm.

<sup>2</sup>  It so happens that the date of 1664—the year of his death—appears in the manuscript data on up to three occasions. This has been considered as a possible error, and that it is in fact a reference to the year 1646 (Mozas Moreno 2018, pp. 72–73).

<sup>3</sup>  Gathering of geographical and historical data requested from the parish priests of the different municipalities of the diocese in the form of a list of ordered and systematised questions.

<sup>4</sup>  As early as the 2nd century AD, Ptolemy, in his Almagest, distinguishes between geography, which study regions and their general features, and chorography. The object of the latter was to deal with particularities, down to the smallest hamlets. As a genre, it is almost impossible to separate from the histories of the cities, focused as it is on describing the smallest details and painting a true portrait of them, pointing out everything noteworthy "such as buildings, houses, towers, walls..." (Apianus 1548).

<sup>5</sup>  A defensive tower detached from the curtain wall and connected to it by a bridge or an arcade.

<sup>6</sup>  This humanist, historian, and archaeologist from Cordoba persuaded Philip II of Spain to order the production of *Relaciones sobre la historia y topografía de los pueblos de España*, a compilation based on answers to questionnaires designed by him in which toponymic, archaeological, historical, and ecclesiastical data were requested from various collaborators who were familiar with monuments, inscriptions, and antiquities. This documentation has been preserved in eight volumes that provide an X-ray of Spain at the time.

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
