# Peer review of "Andalusi Defensive Architecture through Martín de Ximena Jurado’s Drawings (Mid-17th Century)"

_arts_

Round 1
Reviewer 1 Report
The work is very well done and of great scientific interest, with an original and accurate approach. The main issue that arises, and needs to be revised, concerns the transcription of Arabic words. The author(s) sometimes use a popular Spanish system, for example, in lines 214, 217, we see: "Betiz Aben Habuz", "Aben-Zayda." Sometimes a better system without diacritical marks is used, as in lines 720, 724, for example: Banu Bayila, Yusuf ibn Nasr. However, in other cases, the paper follows a correct scientific transcription system, as seen in lines 293, 294, 295, 301; for instance: the kūra of Ŷaīyān, Tušš, ḥiṣn, al-Bayyāsī. The latter should be the system followed throughout the text, so non-academic transcriptions should be avoided, and the same transcription system should always be used.
Other minor issues are the followings:
- In the paragraph ending at line 221, there is a lack of citations for the sources from which the information is derived.
- In lines 60 and 61, it is mentioned that the Kingdom of Jaen had the highest density of medieval fortifications in Europe, but no arguments or sources are provided to support this statement. If this sentence is not reinforced, it is considered a hyperbole that should be removed from the text.
Author Response
The Arabic words are in the same transcription system, with diacritical marks:
Betiz Aben Habuz => Bādīs bin Ḥabūs
Aben-Zayda => Qal‘at Banī Sa’īd, the castle of Benzayde or Abenzaide for the Christian
Banu Bayila => Banū Bāhilah
Muhammad ibn Yusuf ibn Nasr => Muḥammad b. Yūsuf b. Naṣr
Other words in Arabic also have been revised:
ibn Hafsun => Umar ibn Ḥafṣūn
Ibn Mardanish => Ibn Mardanīsh
Yusuf I => Yūsuf I
al-Adil => al-‘Ādil
Bury al-Hamma => Burŷ al-Hammam
al-Hakan II => al-Ḥakam II
Qal’at Aryuna => Qal‘at Arŷūna
ibn al-Hamar => Ibn al-Aḥmar
Muhammad I => Muḥammad I
Banu Nasr => Banū Naṣr
Banu al-Ahmar => Banū l-Aḥmar
Banu Asqilula => Banū Ašqīlūla
Other minor issues are the followings:
In the paragraph ending at line 221, there is a lack of citations for the sources from which the information is derived.
(Eslava Galán 1999: 365-372)
(Eslava Galán 1999) Eslava Galán, Juan. 1999. Los castillos de Jaén. Armilla (Granada): Ediciones Osuna.
In lines 60 and 61, it is mentioned that the Kingdom of Jaen had the highest density of medieval fortifications in Europe, but no arguments or sources are provided to support this statement. If this sentence is not reinforced, it is considered a hyperbole that should be removed from the text.
(Cerezo Moreno and Eslava Galán 1989: 8; Eslava Galán 1999: 17)
(Cerezo Moreno and Eslava Galán 1989) Cerezo Moreno, Francisco, and Juan Eslava Galán. 1989. Castillos y atalayas del Reino de Jaén. Jaén: Riquelme y Vargas Ediciones.
(Eslava Galán 1999) Eslava Galán, Juan. 1999. Los castillos de Jaén. Armilla (Granada): Ediciones Osuna.
Reviewer 2 Report
It is an excellent article. For first time, the complete collection of drawings of fortifications in the Old Kingdom of Granada that were drawn by Martin Ximena in the mid-17th century is published, analyzed and valued. The article provides abundant local data that is essential to understand the documentary value of these drawings on Islamic fortifications in Andalusia. The representation system was very elementary, but coherent and easily understandable. From now on this collection of drawings must be considered very relevant in the European 17th century.
To try to improve the publication, some small minor corrections indicated below should be considered.
-In the summary (line 11) it says that the former territory of the Kingdom of Jaén constitutes the area of the Iberian Peninsula with the highest density of medieval fortifications; and then on line 61 it says: has the highest density of European medieval fortifications. The bibliographical source from which this data comes must be cited.
-Most of Martin Ximena's contributed drawings are in the National Library of Spain. However, in some of them another publication from 1996 is cited: the location of the original should also be cited, if possible: fig. 4b; fig. 9a and 9b, fig.16, fig. 25b.
-In addition, the source of the following figures should be indicated: 10, 11, 12, 13, 14, 15, 16.
-There is an error in line 587. The title must be 2.1.6. Andújar (not Alcalá la Real)
-In section 2.2 (line 848 and following) it is strange to find subsections without any text; surely it is better to include all the images in the same place, but without subsections
-In conclusions, is mentioned Pier María Baldi, and a publication from 1933 is cited in line 1203 (Sánchez Rivero and Mariutti de Sánchez Rivero 1933). There is a later, more extensive and complete publication on Baldi (El viaje a Compostela de Cosme III de Médicis, 2004; ISBN: 84-453-3892-7), which should replace this bibliographical citation: https://dialnet.unirioja.es/servlet/libro?codigo=831855
-In conclusions, are mentioned Leonardo da Vinci and the origins of perspective (line 1213; and line 1229). On this subject, which is much more complex and extensive, the following books should be cited:
Gentil Baldrich, José María: Sobre la supuesta perspectiva antigua y algunas consecuencias modernas. Universidad de Sevilla, Instituto Universitario de Arquitectura y Ciencias de la Construcción, Sevilla, 2011. ISBN 978-84-472-1402-0
Arévalo Rodríguez, Federico: La representación de la ciudad en el Renacimiento. Levantamiento urbano y territorial. Fundación Caja de Arquitectos, Barcelona, 2003. 84-932542-6-6
Author Response
It has been cited the location of fig. 4b; fig. 9a and 9b, fig. 15, fig.16 and fig. 25b (Arjona Municipal Archive), as well as the source for fig. 10, fig. 11, fig. 12, fig. 13 and fig. 14 (Centro Documental y Biblioteca del Instituto de Estudios Giennenses, Diputación Provincial de Jaén, sign. MAP-B 201-1 to 5).
It has been corrected "2.1.6. Andújar".
The subsections have been removed in section 2.2.
In the conclusions, related to Pier María Baldi it has been cited:
(Neira Cruz 2004) Neira Cruz, Xosé Antonio (dir.). El viaje a Compostela de Cosme III de Médicis. Catálogo de la exposición celebrada en el Museo Diocesano, Santiago de Compostela, 15 octubre 2004-17 enero 2005. Santiago de Compostela: Xunta de Galicia, Consellería de Cultura, Comunicación Social e Turismo. S.A. de Xestión do Plan Xacobeo.
Where is mentioned Leonardo da Vinci and the origins of perspective, have been cited:
(Gentil Baldrich 2011) Gentil Baldrich, José María. 2011. Sobre la supuesta perspectiva antigua y algunas consecuencias modernas. Sevilla: Universidad de Sevilla, Instituto Universitario de Arquitectura y Ciencias de la Construcción.
(Arévalo Rodríguez 2003) Arévalo Rodríguez, Federico. 2003. La representación de la ciudad en el Renacimiento. Levantamiento urbano y territorial. Colección Arquithesis, núm. 13. Barcelona: Fundación Caja de Arquitectos.
As this subject is much more complex and extensive, this section has been completed with the following paragraphs:
"By the first third of the 15th century the Italian artistic avant-garde had managed to establish a few rules for the depiction of the drawing space. This trajectory can be traced back to the end of the 13th century, when Giotto di Bondone (1266-1337) developed an early intuitive drawing style, solidly three-dimensional with accurate depiction of perspectives with one and two vanishing points. He was followed by great masters of the 14th and 15th centuries such as Ambrogio Lorenzetti (1290-1348), Bernardo Daddi (ca. 1312-1348), Andrea di Bonaiuto (ca. 1346-1379), Filippo Brunelleschi (1377-1446), Donato di Niccolò di Betto Bardi –Donatello– (ca. 1386-1466), Paolo Ucello (1397-1475), Stefano di Giovanni –Il Sassetta– (1392-ca. 1451), Tommaso di Ser Giovanni di Simone –Masaccio– (1401-1428), Lorenzo Ghiberti (1378-1455), Antonio Pollaiulo (1429-1498), Andrea del Verrochio (ca. 1435-1488), Domenico Ghirlandaio (1449-1494), Sandro Boticelli (1445-1505), Francesco Raibolini –Francesco Francia– (1447-1517) or Antonello da Messina (ca. 1430-1479).
Although without following a complete systematisation, these artists had already adopted the single vanishing point for parallels, as would be the case in Flanders shortly afterwards. In the north of the continent, outstanding figures emerged such as Jan Van Eyck (1395-1441), who is considered one of the forerunners of perspectival methods. Later on, Petrus Christus (1410/1415-1475/1476) developed a unitary conception of space by simplifying the models of his predecessor, which enabled him to empirically discover the basic rule of linear perspective by constructing his compositions with a single vanishing point. Dirk Bouts (1415-1475) also made use of this principle in some of his works.
It is also worth noting the development of the representation of architectural spaces and volumes from the latter part of the 13th century in the genre known as Book of Hours or Horarium. These manuscripts are characterised by their extensive illumination and are a source of iconography of medieval Christianity. Architectural volumes resolved by means of conical and axonometric perspectives often form part of the composition. In some of these works, the powerful presence of castles, towers and walled cities sometimes stands out in the background of the images.
Against this background, the standard of graphic representation that spread began to make use of conical perspective. Its origin cannot be established in ancient Hellenistic and Roman paintings, nor in the studies of Euclidean optics – rather it started to develop in the Middle Ages. It came from a set of practical methods elaborated from the 13th century by European painters, who gradually began to adopt the convergence of parallels to a centred vanishing point of the composition and, even to two vanishing points in some elements of the scene (Grayson 1964; Panofsky 2003; Neira Cruz 2004; Gentil Baldrich 2011; Ramón-Laca Menéndez de Luarca 2020).
Leon Battista Alberti (1404-1472) is considered to be the author of the first treatise written in Latin on conic perspective, found in his work De pictura (Rome, ca. 1435). It was the first of a trilogy of treatises, together with De re aedificatoria (ca. 1450) and De statua (ca. 1462), and was widely disseminated within the humanities. In this work Alberti asserted: “No one of understanding will deny that no painted thing can be seen to resemble reality unless a certain reason is employed”. Since Alberti’s work, the theory of graphic representation in perspective developed enormously in with works such as those of Leonardo da Vinci (1452-1519), who, although he did not publish any specific work on the subject, on fol. 42r of his manuscript A, sketches the construction of a method of perspective. Also noteworthy are the treatises by Italians such as Sebastiano Piero della Francesca (De Prospettiva Pigendi, ca. 1475), Ponponio Guarico (De sculptura:… De perspectiva, 1504), Fra Luca Pacioli (De divina proportione, 1509), Monsignor Danielle Barbaro (La Prattica della Perspettiva, 1569), the also cleric Egnatio Danti (Le due regole della prospettiva prattica, 1583, with the compilation of writings of M. Jacopo Barozzi da Vignola), Lorenzo Sirigatti (La pratica di prospettiva, 1596) or Sebastiano Serlio (Il secondo libro d’Architettura, 1545, dedicated to perspective and scenography).
In northern and central Europe, perspectival codes had become extraordinarily widespread, and had acquired a greater normative rigour to Italian models. Albert Dürer published Underweysung der messung in Nuremberg in 1525, which influenced many other German authors, with geometry being more important than perspective. Dürer was keen to depict perspective machines in his work, two in the 1525 edition, increasing to four in the 1538 edition. Other treatises that would have been influenced by him were Hieronymus Rodler (Underweysung der kunts des Messens dem Zirckel Richtscheidt oder Linial, 1531), Ulrich Kern (Eyn new kuntsliche wolgegründt Visierbuch, 1531), Walther Hermann Ryff (Der neuen Perspectiva, 1547), Heinrich Lautensack (Des Circkels unnd Richtscheyts, auch der Perspectiva und Proportion der Menschen und Rosse, kurtze, doch gründtliche underweisung deß rechten gebrauchs, 1564), Wenzel Jamnitzer (Perspectiva corporum regularium, 1568), Lorenz Stoer (Geometria et perspectiva, 1567), or Hans Lencker (Perspectiva, 1571).
From the beginning of the 16th century, also France saw a stream of treatises emerged, which began with the publication in 1505 of the treatise De artificialis perspectiva, by the canon of Toul (France) Jean Pélerin “le Viateur”. It was written in Latin and French to “proceed in representing and artificially figuring things seen or conceived”. In this work, Pélerin developed the rule of “tiers point” or distance points, to represent oblique perspectives with two vanishing points on the horizon line. In his work the drawings are more important than the text, taking from geometry only those lines that are necessary for the representation. Following this theory of perspective, to which Italy and northern Europe remained practically unfamiliar for much of that century, other Frenchmen such as Jean Cousin (Livre de Perspective, 1560) and Jacques Androuet du Cerceau (Leçons de perspective positive, together with Les plus excellents bastiments de France, 1576) also published treatises. In them, the conic language began to be transgressed. In some illustrations, an axonometric pre-language appeared, representing parallel straight lines without vanishing point. That would later give rise to the “military perspective”, with the possibility of measuring in visual representations, given the geometric complexity of the layout of the conic. In military engineering, the design and construction of fortifications required them to be rationally measured, which gave rise to dimensionally controllable perspective systems. Another cleric, Father De Breuil, also referred to military perspective in his work Perspective prattique par un religieux de la compagnie de Jesus (1639).
This treatise was published in the same year in which Ximena Jurado may have begun to write his Ms, 1180 B.N. However, despite this vast tradition that preceded him by more than two centuries, he seems not to have been familiar with these avant-garde movements nor the gradual evolution of perspective since the 13th century in paintings and reliefs with Christian themes. In addition, among these treatises there are a number of publications by religious authors. But Ximena Jurado’s depictions do not systematically reflect any of the perspectival theories developed up to that time, currents to which Spain had not remained on the fringes. In this respect, one might point to Cristóbal de Rojas’s treatise Teorica y practica de fortificacion, conforme las medidas y defensas destos tiempos: repartida en tres partes (1598), which appeared in Madrid several decades before Ximena Jurado’s drawings of castles, towers and walled cities."
Reviewer 3 Report
No comments
There are some words and concepts translated into English which I think are not correct. For instance, "antique dealer"
There are also some strange grammar structures: "Writing his works, he followed the chronological compilation of data of historical interest, mainly on artefacts from archaeological or documentary sites, which he interpreted through a wide and varied bibliography"
Author Response
There are some words and concepts translated into English which are not correct. For instance, "antique dealer". It has been changed for “antiquarian”.
There are also some strange grammar structures: "Writing his works, he followed the chronological compilation of data of historical interest, mainly on artefacts from archaeological or documentary sites, which he interpreted through a wide and varied bibliography". It has also been changed for: “He followed the chronological compilation of data of historical interest, which he interpreted through a wide and varied bibliography.”
Reviewer 4 Report
A few typos detected. E.g. 339 "she"; 453 cut "what"; 548 "he".
829: A rasif is an ordered, built mass of, e.g., masonry or brick; and hence, e.g., quay or especially today a (train) platform. I've not come across the term to mean a stone road. It's possible.
Author Response
The typos detected have been corrected:
Line 339: "she" => he
Line 453: "what" has been removed
Line 549: "he gateway has direct access " => The gateway has direct access
Line 829: A rasif is an ordered, built mass of, e.g., masonry or brick; and hence, e.g., quay or especially today a (train) platform. It has been removed "stone road".
“el Arrecife” (from Arabic al-raṣīf, stone road) => “el Arrecife” (from Arabic al-raṣīf)
-------------------
Franco-Sánchez, F., (2017). La toponimia árabe de los espacios viales y los espacios defensivos en la península Ibérica. In Carvalho, C., Planelles Iváñez, M., Sandakova, E. & Aragón Cobo, M. De la langue à l’expression : le parcours de l’expérience discursive. Hommage à Marina Aragón Cobo (pp. 167-190). Alicante: Universitat d’Alacant.
(p. 154) “El término castellano “arrecife” en su sentido originario era un arabismo derivado del étimo al-raṣīf cuyo significado es ‘el empedrado’, y de ahí ‘la calzada’ (...). Leopoldo Torres Balbás le ha dedicado un estudio pormenorizado al tratar sobre la Vía Augusta, recogiendo del Vocabulista in Arabico de Pedro de Alcalá el que los musulmanes denominaban “arrecife” a todo camino enlosado o empedrado, como lo estaban las principales calzadas romanas y entre ellas la Vía Augusta (Torres Balbás 1959: 448; Pocklington 1990: 134)”.
Torres Balbás, L., (1959). “La Vía Augusta y el arrecife musulmán”, Al-Andalus, 24, pp. 441-448.
Pocklington, R., (1990). “La huerta y campo de Murcia en los siglos VIII-IX vistos a través de su toponimia”. In: Pocklington, R., Estudios toponímicos en torno a los orígenes de Murcia, Murcia, Academia Alfonso X el Sabio, pp. 111-146.